# Fluorescent fatty acid conjugates for live cell imaging of peroxisomes

Daria Korotkova[1], Anya Borisyuk[1], Anthony Guihur [2], Manon Bardyn [3], Fabien Kuttler [3], Luc Reymond [3], Milena Schuhmacher [4] & Triana Amen [1,5] ✉

Peroxisomes are eukaryotic organelles that are essential for multiple metabolic pathways, including fatty acid oxidation, degradation of amino acids, and biosynthesis of ether lipids. Consequently, peroxisome dysfunction leads to pediatric-onset neurodegenerative conditions, including Peroxisome Biogenesis Disorders (PBD). Due to the dynamic, tissue-specific, and context-dependent nature of their biogenesis and function, live cell imaging of peroxisomes is essential for studying peroxisome regulation, as well as for the diagnosis of PBD-linked abnormalities. However, the peroxisomal imaging toolkit is lacking in many respects, with no reporters for substrate import, nor cell-permeable probes that could stain dysfunctional peroxisomes. Here we report that the BODIPY-C12 fluorescent fatty acid probe stains functional and dysfunctional peroxisomes in live mammalian cells. We then go on to improve BODIPY-C12, generating peroxisome-specific reagents, PeroxiSPY650 and PeroxiSPY555. These probes combine high peroxisome specificity, bright fluorescence in the red and far-red spectrum, and fast non-cytotoxic staining, making them ideal tools for live cell, whole organism, or tissue imaging of peroxisomes. Finally, we demonstrate that PeroxiSPY enables diagnosis of peroxisome abnormalities in the PBD CRISPR/Cas9 cell models and patient-derived cell lines.

Tracking the morphology, localization, and function of cellular organelles in live cells with fluorescent molecules has transformed our understanding of organelle dynamics and revealed previously unknown facets of numerous pathologies (e.g., mitochondria redox potential-dependent or lysosome pH dependent dyes)[1–5]. Peroxisomes, eukaryotic single membrane-enclosed compartments that are critical for lipid metabolism and redox homeostasis, are typically visualized in live cells with peroxisome-targeting sequence fusions[6–9]. Though this approach provides excellent specificity, it is often limited to tracking functional peroxisomes and requires the introduction of genetic material into the cell. Existing peroxisome

tracking approaches are, therefore, inadequate for investigating peroxisome dysfunction and peroxisome biogenesis abnormalities, in particular those that may occur in patient samples. In addition, there is a lack of selective methods for non-invasive labeling of peroxisomes in live cells and tissues, though such tools would be invaluable for research and diagnostic applications. Only a few live cell imaging probes have so far been developed, including a BODIPY-based probe that only labels peroxisomes in plants (but not in mammalian cells)[10,11], and a probe that consists of the peroxisome targeting sequence peptide fused to a fluorescent probe, rendering it dependent on functional protein import into peroxisomes[12]. It is

[1]Global Health Institute, Faculty of Life Sciences, Ecole Polytechnique Fédérale de Lausanne (EPFL), Lausanne, Switzerland. [2]Department of Plant Molecular Biology, Faculty of Biology and Medicine, University of Lausanne, Lausanne, Switzerland. [3]Biomolecular Screening Facility, Ecole Polytechnique Fédérale de Lausanne (EPFL), Lausanne, Switzerland. [4]Institute of Bioengineering, Faculty of Life Sciences, Ecole Polytechnique Fédérale de Lausanne (EPFL), Lausanne, Switzerland. [5]School of Biological Sciences, University of Southampton, Southampton, UK. ✉e-mail: t.amen@soton.ac.uk

therefore essential to develop techniques to visualize functional as well as aberrant peroxisomes.

Peroxisomes continuously import and export metabolites and substrates, such as very long chained fatty acids for beta oxidation[6]. This activity requires a series of transporters spanning the peroxisomal membrane[13,14]. In humans, the prominent transporters are ATP-binding cassette (ABC) transporters (ABCD1, ABCD2, and PMP70 (ABCD3)), and a transmembrane channel for small metabolites, PXMP2[14–19]. The importance of substrate trafficking for peroxisomal function is emphasized by two observations: PMP70 is the most abundant peroxisomal transmembrane protein, and an ABCD1 transporter mutation is the most common genetic cause of peroxisome biogenesis disorder (PBD), an X-linked adrenoleukodystrophy, in young children[16,17,20,21].

Here we report an application of BODIPY-C12 as a peroxisome dye that can be used to track peroxisomes in live animal cells and live Zebrafish embryos. BODIPY-C12 accumulates in functional peroxisomes in a PMP70-dependent manner, but also stains import-deficient peroxisomes (albeit at a markedly slower rate). We further developed a set of probes, PeroxiSPY650 and PeroxiSPY555, with improved peroxisome specificity and a fluorescence profile that is well-suited for live cell, animal, and deep tissue imaging. We also show that the rate of dye import can be used as a reporter of peroxisome function. Finally, we demonstrate that BODIPY-C12 and PeroxiSPY probes can be used to detect peroxisome abnormalities in iPSC cells derived from a PBD

patient and that PeroxiSPY can detect peroxisome deficiency in patient fibroblasts lacking functional PEX3.

## Results

### Fluorescent fatty acid BODIPY-C12 accumulates in peroxisomes in live mammalian cells

BODIPY-C12 is a fluorescent fatty acid that is widely used to track lipids that accumulate in lipid droplets and mitochondria[22–25]. We identified unspecified BODIPY-C12-positive "specks" in live mammalian cells that appeared before lipid droplet staining (Fig. 1a)[26]. The BODIPY fluorophore is itself a lipophilic molecule that is commonly used as a dye to stain lipid droplets[27]. However, the BODIPY-C12 specks did not co-localize with Lipid Droplets (Fig. 1a). Other BODIPY conjugates (BODIPY-methyl ester, and Cholesterol-BODIPY), on the other hand, did stain Lipid Droplets and did not produce a "speck" pattern (Supplementary Fig. 1a, b). Nor did lysosome staining with NIR-633 co-localize with the BODIPY-C12 staining specks (Supplementary Fig. 1c). Since peroxisomes process several types of fatty acid lipids, including branched chain and very long chain fatty acids[6], we reasoned that a conjugated fatty acid probe such as BODIPY-C12 may accumulate in peroxisomes. We visualized peroxisomes in live cells using the lumenal peroxisomal import marker GFP-SKL. BODIPY-C12 fluorescent specks indeed co-localized fully with peroxisomes (Fig. 1b and Supplementary Fig. 1d). To definitively prove peroxisome localization, we created

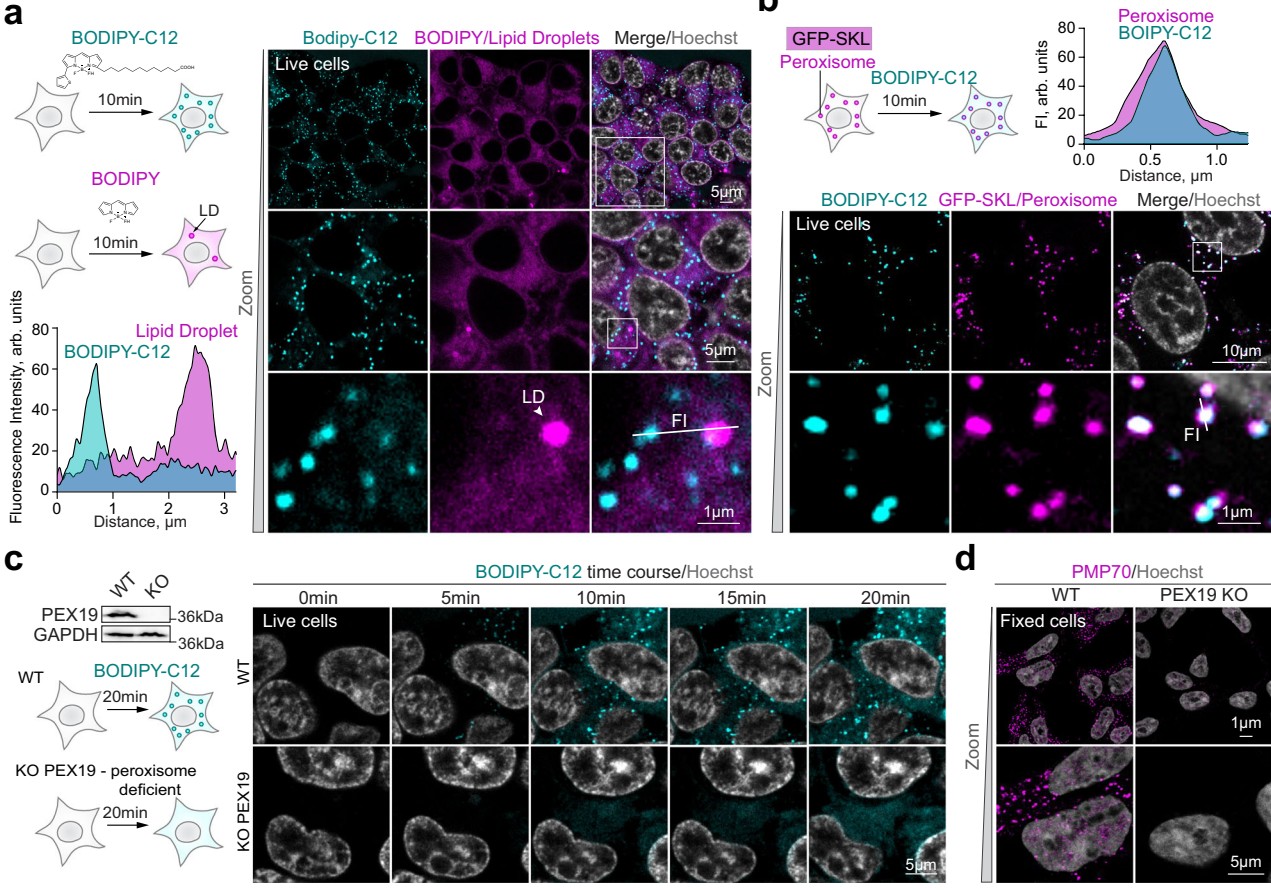

**Fig. 1 | Conjugated fluorescent fatty acid BODIPY-C12 accumulates in peroxisomes in live cells. a** Confocal microscopy of live HEK293T cells stained with BODIPY-C12 (1 µg/ml, cyan) and BODIPY (1 µg/ml, magenta) for 10 min. Nuclei were stained with Hoechst (10 µg/ml). A representative intensity profile is shown, Pearson Correlation Coefficient (PCC) = 0.12, scale bars − 5 and 1 µm. LD – lipid droplet, FI – fluorescence intensity. **b** Confocal microscopy of live HEK293T cells expressing GFP-SKL(magenta), stained with BODIPY-C12 (1 µg/ml, cyan) for 10 min. Nuclei were stained with Hoechst (10 µg/ml). A representative intensity profile is shown, PCC = 0.86, scale bars − 10 and 1 µm. **c** Confocal microscopy of live CRISPR/Cas9 PEX19 KO and WT HEK293T cells during BODIPY-C12 (1 µg/ml) staining, scale bar − 5 µm. Nuclei were stained with Hoechst (10 µg/ml). Western blot shows PEX19 levels in WT and PEX19 KO cells. **d** Confocal microscopy of fixed CRISPR/Cas9 PEX19 KO and WT HEK293T cells stained with PMP70 antibody, scale bar − 5 µm. Nuclei were stained with Hoechst (10 µg/ml).

CRISPR/Cas9 *PEX19* KO cells that lack peroxisomal compartments entirely[28] (Fig. 1c, d). BODIPY-C12 accumulated in peroxisomes in WT but not *PEX19* KO cells (Fig. 1c, d). To account for possible cell line bias, we repeated the peroxisome-BODIPY-C12 co-localization experiment in an independent human cell line. We observed similar peroxisome staining in human U2OS cells (Supplementary Fig. 1e).

## Synthesis of a peroxisome specific dye PeroxiSPY

Encouraged by the possibilities that tractable probes can provide for the study of peroxisome biology, we decided to iterate on the chemistry of BODIPY-C12 to make a probe for peroxisomes with substantially higher specificity. We hypothesized that a longer fatty acid would preferentially accumulate in peroxisomes and that a non-lipophilic dye will not show any lipid droplet staining. We chose silicone rhodamine (SiR) that is a far-red, photostable, and cell-permeable dye suitable for super-resolution microscopy[29], and the red Map555 dye developed for live cell nanoscopy[30]. We conjugated the dyes to C16, C18, or C20 fatty acids bearing a terminal alkyne using copper-mediated click chemistry (Supplementary Figs. 2–4). Indeed, cell-permeable dyes (SiR-C16, SiR-C18, SiR-C20, MaP555-C18, and MaP555-C20) stained peroxisomal compartments (Supplementary Fig. 5a) exclusively, with SiR-C20 and MaP555-C20 demonstrating the highest peroxisome to cytoplasm ratio (Supplementary Fig. 5b) We next measured potential effects on cell growth and viability. At the concentrations used for live cell imaging (0.1–1 μM), the dyes were not toxic and did not affect the cell growth, except for SiR-C18 effect that reduced cell growth at 1–10 μM concentrations (Supplementary Fig. 6a–c). Interestingly, high (10 μM) concentrations of SiR-C20 were toxic for *PEX19* KO cells but not WT cells after 24 h of incubation (Supplementary Fig. 6a). A similar phenomenon was previously demonstrated for peroxisome-processed pyrene fatty acid conjugates[31]. We further focused on the best performing dyes – SiR-C20 and SPY555-C20, which we will refer to as PeroxiSPY650 and PeroxiSPY555, respectively. A side-by-side comparison with BODIPY-C12 and a BODIPY-FL-C12(green) showed that BODIPY-based dyes have comparable brightness in peroxisomes (Fig. 2a, Supplementary Fig. 5b). We then assessed the staining specificity, by co-staining Lipid Droplets using either Nile Red or BODIPY (non-conjugated) (Fig. 2b–e). As expected, both BODIPY conjugates accumulated in Lipid Droplets in addition to peroxisomes after 30 min of incubation (Fig. 2d, e). PeroxiSPY dyes, however, exhibited uniquely peroxisome-specific staining throughout the experiment that was confirmed using *PEX19* KO cells that lack peroxisomes and real-time imaging of peroxisomes (Fig. 2b, c and Supplementary Movie 1). We then measured the displacement of BODIPY-C12 by PeroxiSPY650 by incubating cells with a mix of the dyes, gradually increasing the concentration (Fig. 2f). PeroxiSPY demonstrated superior retention in peroxisomes, significantly displacing BODIPY-C12 from peroxisomes (half of BODIPY-C12 fluorescence is lost with 1:1.2uM BODIPY-C12 to PeroxiSPY650 ratio), while BODIPY-C12 only displaced PeroxiSPY650 at a 10-times the concentration (Fig. 2g). Next, we examined the mechanisms of dye association with peroxisomes.

## Peroxisome staining by BODIPY-C12 is modulated by peroxisomal function

Peroxisomes are usually visualized with the import marker GFP-SKL, which is imported into peroxisomes by the PEX5 import receptor[32,33]. In the absence of PEX5, GFP-SKL remains cytoplasmic (Fig. 3a)[34]. BODIPY-C12 stained peroxisomes independently of peroxisomal function in *PEX5* KO (Fig. 3a). We noticed, however, that staining in PEX5 KO cells is transient, briefly disappearing after 15 min. We therefore investigated the dynamics of the BODIPY-C12 accumulation in the WT, *PEX5* KO, and stably complemented *PEX5* KO cells with a PEX5L isoform[35] (Fig. 3b, c). WT cell staining lasted significantly longer than that of cells with dysfunctional peroxisomes, and the staining was restored upon complementation with overexpressed PEX5L (Fig. 3d, e). A similar staining pattern (a transient accumulation in *PEX5* KO peroxisomes) was observed for the PeroxiSPY dyes (Fig. 3f–h and Supplementary Fig. 5c). We also noticed that the timing of BODIPY-C12 staining in WT cells inversely depended on the dye concentration (Supplementary Fig. 5d): higher concentrations led to an overload of peroxisomal staining capacity and spilled over to other cellular membranes, while lower concentrations of the BODIPY conjugate resulted in more specific staining (100% of cells had peroxisome staining after an hour incubation). PeroxiSPY staining was not dependent on the range of concentrations tested (Supplementary Fig. 5d, e).

Fatty acids are imported into peroxisomes for fatty acid oxidation through ABCD1-3 transporters while small molecules pass through the PXMP2 channel[19]. BODIPY-C12 and PeroxiSPY dyes are above the 300 Da free diffusion cut-off of the PXMP2 channel[19], thus we posited that it may require ABC transporters to enter peroxisomes. We created a CRISPR/Cas9 KO of the most abundant ABC transporter on peroxisomal membranes - PMP70[15,16] (Fig. 4a and Supplementary Fig. 1f), which abolished BODIPY-C12 peroxisome staining (Fig. 4a, b), and significantly reduced the PeroxiSPY staining (Fig. 4c–e).

Finally, we asked whether the dyes were incorporated through a peroxisome-ER contact site implicated in lipid trafficking between ER and peroxisomes[36,37]. We used a double KO of *VAPA* and *VAPB*[38] that we find results in a significant defect in peroxisome biogenesis (a reduced number of peroxisomes) (Fig. 4f–h). The dyes incorporated into WT and *VAPA/VAPB* KO peroxisomes, even to a higher extent in the mutant lacking peroxisome-ER VAPB-ACBD4/5 contacts (Fig. 4f–j), indicating that contact sites unlikely mediate the import of the dyes into peroxisomes. These results suggest instead the possibility of substrate-like import by ATP-dependent transporters. Indeed, peroxisomes in fixed cells were stained neither by the BODIPY-C12-based dyes nor by our optimized probes (Fig. 4k).

## Peroxisome staining enables the identification of peroxisome abnormalities in patient cells

To determine whether peroxisome staining could identify peroxisome dysfunction in disease contexts, we visualized peroxisomes in a patient-derived iPSC cell line with a PEX10 compound mutation which reduces the activity and abundance of PEX10 (Fig. 5a). PEX10 regulates peroxisome biogenesis, PEX5 turnover, and impacts protein import into peroxisomes[39–42]. In PEX10 mutant cells, import of GFP-SKL was delayed after 24 h of GFP-SKL expression (Fig. 5b). Stem cell peroxisome staining with BODIPY-C12 was still present after two hours of incubation with a noticeable gradual increase in the cytoplasmic signal. The cytoplasmic signal was also significantly increased in PEX10 mutant cells (Fig. 5c). We next stained iPSCs with PeroxiSPY650 (Fig. 5d). Control cells exhibited significantly brighter peroxisomal signal, the ratio of peroxisomes to cytoplasm fluorescence intensity was decreased in patient cells (Fig. 5d), and peroxisomes in patient cells were more numerous (Fig. 5e). Interestingly, however, the function of peroxisomes measured by very long chain fatty acid oxidation using radioactive 3H-docosanoic (C22) acid was not different in control and patient stem cells (Fig. 5f). We next examined *PEX3*-deficient patient fibroblasts that lack peroxisomal compartments, which can be restored by transient expression of WT *PEX3* (Fig. 5g)[43,44]. Interestingly, only PeroxiSPY dyes stained peroxisomes in human fibroblasts, clearly showing absence of peroxisomal compartments in *PEX3*-deficient patient fibroblasts, and restored peroxisomes in a complemented patient cell line (Fig. 5g, h and Supplementary Fig. 1g).

## Visualizing peroxisomes in a live Zebrafish embryo

Next, we asked whether tissue and organism staining is possible with our peroxisome dyes. As a model organism, we chose the zebrafish, *Danio rerio*. We stained the embryo (16–22 h post-fertilization (hpf)) with BODIPY-C12 and PeroxiSPY. Dechorionated embryos

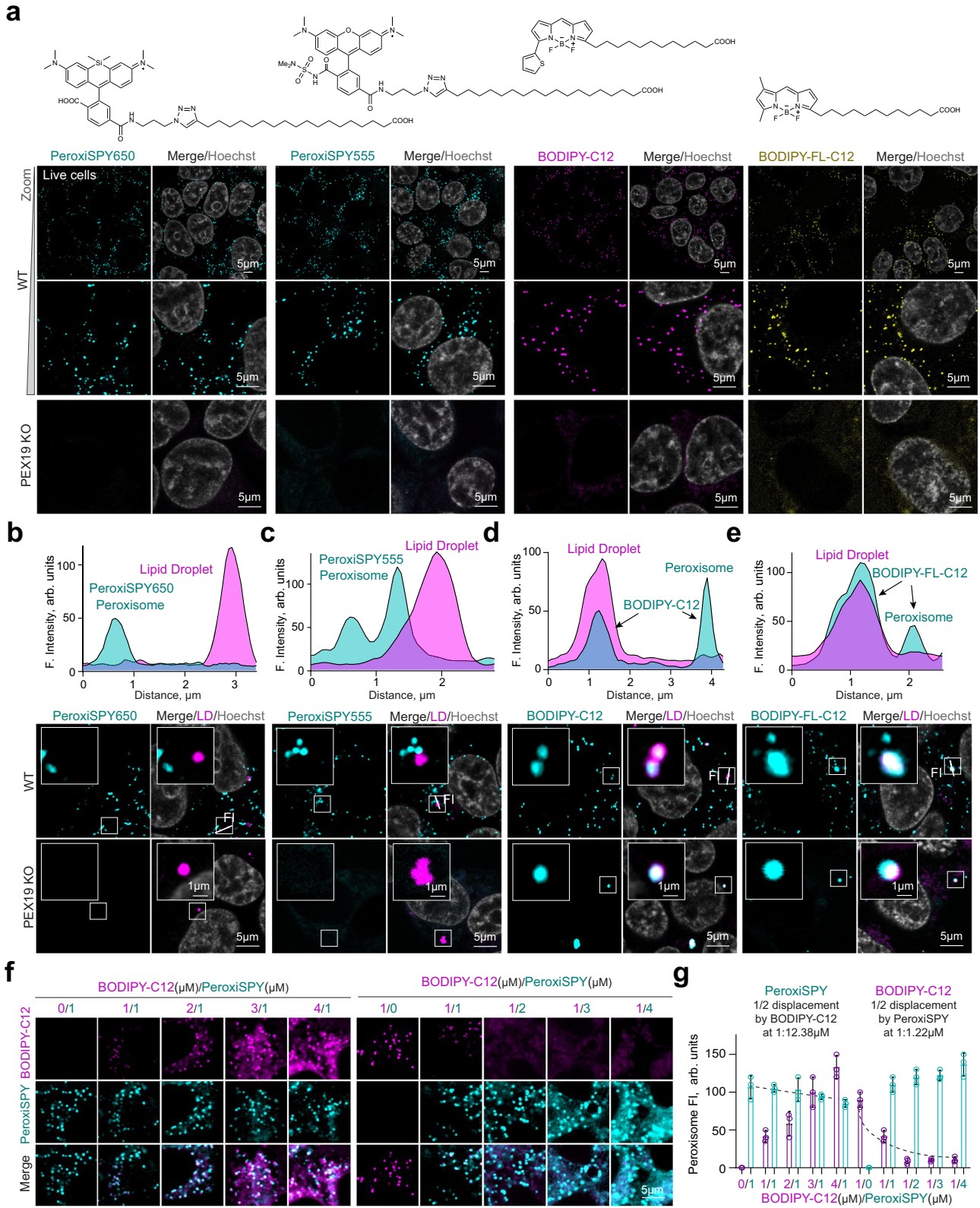

**Fig. 2 | Peroxisome-specific dye PeroxiSPY. a** Confocal microscopy of live WT and PEX19 KO HEK293T cells stained with PeroxiSPY650 (1 µM), PeroxiSPY555 (1 µM), BODIPY-C12 (1 µg/ml), or BODIPY-FL-C12 (green) (1 µg/ml) for 10 min. Nuclei were stained with Hoechst (10 µg/ml), scale bar − 5 µm. Probe structures are shown. **b–e** Confocal microscopy of live WT and PEX19 KO HEK293T cells stained with (**b**) PeroxiSPY650 (1 µM, cyan) and Nile Red(magenta), (**c**) PeroxiSPY555 (1 µM, cyan) and BODIPY(magenta), (**d**) BODIPY-C12 (cyan) and BODIPY (magenta) (**e**) BODIPY-FL-C12 (cyan) and Nile Red(magenta) (1 µg/ml) for 10 min. Nuclei were stained with

Hoechst (10 µg/ml), scale bars − 1 and 5 µm. Representative fluorescence intensity profiles are shown. **f, g** Confocal microscopy of live WT HEK293T cells stained simultaneously with PeroxiSPY650(cyan) and BODIPY-C12(magenta) of indicated concentrations (µM) for 10 min. scale bar − 5 µm. Quantification shows the peroxisome fluorescence intensity of each dye, mean ± SEM, $N = 3$ replicas of an average peroxisome fluorescence intensity pooled from > 300 peroxisomes per replica.

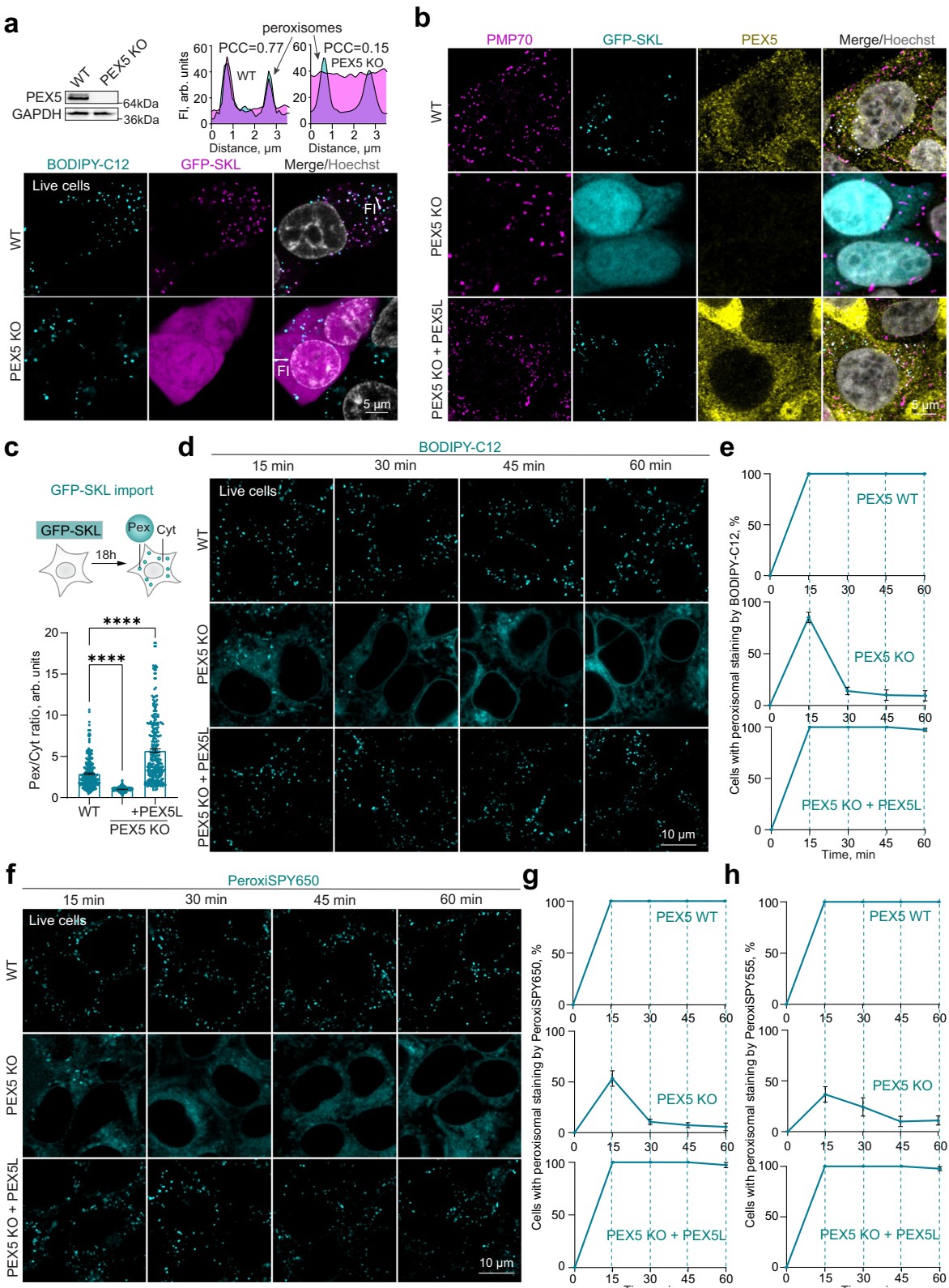

incorporated BODIPY-C12 dissolved in fish water, exhibiting a similar staining pattern to what we observed in cell culture (Supplementary Fig. 7a). Zebrafish embryos are maintained in fish water, which may affect fatty acid solubility, however, addition of albumin to the media in an attempt to improve fatty acid solubility, significantly reduced dye incorporation (Supplementary Fig. 8a). Next, to confirm peroxisomal localization in live embryos we injected single-cell-stage embryos with

mRNA encoding the peroxisome targeted GFP-SKL import marker or control GFP, producing robust staining of peroxisomal compartments (Supplementary Fig. 7b). Incubating live 16–18 hpf embryos with BODIPY-C12 for 10 min showed that the dye indeed incorporates into peroxisomes in live animals (Supplementary Figs. 7c and 8b). The PeroxiSPY dye also stained peroxisomes labeled with GFP-SKL in Zebrafish embryos, however, it required a longer incubation time to

**Fig. 3 | Peroxisome staining by BODIPY-C12 and PeroxiSPY is modulated by peroxisomal function. a** Confocal microscopy of live WT and PEX5 KO HEK293T cells expressing GFP-SKL (magenta) and stained with BODIPY-C12 (1 µg/ml, cyan) for 10 min. Nuclei were stained with Hoechst (10 µg/ml). Representative intensity profiles are shown, PCC (WT) = 0.77, PCC (PEX5 KO) = 0.15, scale bar − 5 µm. Western blot shows PEX5 levels in WT and PEX5 KO cells. **b, c** Confocal microscopy of fixed WT, PEX5 KO, and complemented PEX5 KO PEX5L HEK293T cells expressing GFP-SKL, stained with PMP70 and PEX5 antibodies. Nuclei were stained with Hoechst (10 µg/ml), scale bar − 5 µm. **c** Quantification shows the ratio of peroxisome to cytoplasm fluorescence intensity of GFP-SKL

marker, mean ± SEM, **** - $p < 0.0001$, $N = 300$ cells, Kruskal-Wallis test. **d, e** Confocal microscopy of live WT, PEX5 KO, and complemented PEX5 KO PEX5L HEK293T cells, stained with BODIPY-C12 (1 µM) for the indicated amounts of time, scale bar − 10 µm. Graphs show the ratio of cells with identifiable peroxisomes, mean ± SEM, $N = 3$ pooled from 300 cells for each time point. **f–h** Confocal microscopy of live WT, PEX5 KO, and complemented PEX5 KO PEX5L HEK293T cells, stained with PeroxiSPY (1 µM) for the indicated amounts of time, scale bar − 10 µm. Graphs show the ratio of cells with identifiable peroxisomes, mean ± SEM, $N = 3$ pooled from 300 cells for each time point.

incorporate into deeper tissue levels than BODIPY-C12 (Supplementary Figs. 7e, f and 8c, d).

## BODIPY-C12 and PeroxiSPY dyes do not stain peroxisomes in *Arabidopsis thaliana*

The BODIPY-based dye for peroxisomes that was previously reported only stains plant cells[10,11]. We, therefore, tested whether BODIPY-C12 and PeroxiSPY dyes could also stain peroxisomes in the model plant *Arabidopsis thaliana*. We utilized an Arabidopsis line expressing GFP fused to the peroxisomal targeting signal SKL integrated into the genome as a marker for peroxisomes[45,46]. Peroxisome staining was analyzed in protoplast suspensions (for studying cellular processes without the interference of the cell wall) and 21-day-old seedling roots. In both assays, neither BODIPY-C12 nor PeroxiSPY650 co-localized with the peroxisomal GFP-SKL marker (Supplementary Fig. 8e–g).

## Discussion

Here we present a fast and specific method to visualize peroxisomes using commercially available reagents in live cells and tissues, including whole Zebrafish embryos, patient-derived fibroblasts, and iPSCs. We have also synthesized highly specific far-red and red photostable peroxisomal dyes that we collectively call PeroxiSPY. These probes provide a complementary approach to existing methods of visualizing peroxisomes.

We provide a model of the probe incorporation mechanism (Fig. 5i). We show that peroxisome staining is dependent on PMP70 making these dyes a useful screening tool for fatty acid import deficiencies. It is possible that the dyes are transferred into peroxisomes through the PMP70 (BODIPY-C12) and other peroxisome membrane transporters (PeroxiSPY) or transporter presence changes the properties of the membranes that allow probe incorporation. Differential expression of transporters in different cell types (e.g., fibroblasts) may in this case determine the specificity of staining: only PeroxiSPY dyes incorporated into peroxisomes in human fibroblasts. In addition, our proof-of-concept data in mammalian cells demonstrate that accumulation of the dyes in peroxisomes depends on peroxisomal function and is sensitive to minor changes in peroxisomal matrix protein import rates (e.g., PEX10 mutation). Additionally, the duration of peroxisomal staining by PeroxiSPY is dependent on the matrix protein import. The absence of protein import in PEX5 KO cells resulted in a reduced staining duration, perhaps due to the dyes binding to certain matrix proteins in the WT peroxisomes. We also observed that peroxisomes have a fixed capacity for absorbing BODIPY-C12 before it leaks into other cellular membranes. This suggests the possibility that peroxisomes protect cells from excess incorporation of unnatural fatty acids as implied previously for Pyrene-C12 conjugated fatty acids[31]. It is possible that the dyes exit peroxisomes through the peroxisome-ER contact sites[36,37]. That explains a stronger staining in the absence of peroxisome-ER VAP-mediated contact sites. However, there also may be a larger peroxisomal capacity to incorporate the dyes due to a significant peroxisome biogenesis disruption in *VAPA/B* KO (Fig. 4i) – resulting in fewer bigger peroxisomes.

We demonstrated live whole fish embryo peroxisomal staining. This proof of concept creates opportunities for non-invasive

organismal and tissue imaging of peroxisomes. Tissues that have increased content of triacylglycerol (neutral lipids) will preferentially incorporate BODIPY-C12 into lipid droplets, making the signal more difficult to interpret, whereas PeroxiSPY resolves this issue with a significant increase in the specificity of PeroxiSPY dyes compared to BODIPY-C12. PeroxiSPY specificity for peroxisomes can also be exploited to deliver a moiety designed to modify peroxisomal function, or to identify its constituents, thus expanding our toolkit for investigating the basic biology of peroxisomes.

The dyes did not stain peroxisomes in the model plant *A. thaliana*. Testing dye performance in evolutionarily divergent organisms highlights potential applications as well as limitations. Plant peroxisomes are functionally different from human peroxisomes, importing different substrates with potential impact on the peroxisomal membrane composition that may explain the lack of peroxisome staining[47] by PeroxiSPY. Further analysis of the factors enabling staining in animal but not in plant peroxisomes could provide insight into peroxisome membrane biology.

Finally, the PeroxiSPY peroxisome-specific staining is highly enabling in a PBD diagnostic setting. We used *PEX19* and *PEX5* knockouts as models of peroxisome biogenesis disorders[48,49], and utilized PBD patient-derived iPSCs and patient-derived fibroblasts[43,44] to investigate the sensitivity of the probe to a decrease in peroxisomal biogenesis and function. Both BODIPY-C12 and PeroxiSPY probes exhibited differential staining of functional and dysfunctional peroxisomes: PeroxiSPY enabled immediate differentiation by quantifying the dye incorporation into peroxisomes, whereas BODIPY-C12 accumulated more outside peroxisomes over time in patient iPSC cells comparing to WT *PEX10* control. PeroxiSPY dyes distinguish between WT and patient-derived fibroblasts, showing the potential of these dyes for fast diagnostics of peroxisomal dysfunctions in vitro and in disease settings.

## Methods
### Ethical statement
The project is designed in full compliance with the guidelines in ethics and responsible research as per UK concordat. The research was approved by the University of Southampton Research Ethics Committee (approval ID90603).

### Cell culture and cell lines
HeLa (gift from Prof. Daniel Kaganovich), U2OS (gift from Prof. Suliana Manley), and HEK293T (Kaganovich Lab) cells were maintained in high glucose DMEM supplemented with 10% fetal bovine serum (FBS), 1% penicillin/streptomycin, at 37 °C/5% CO2. Human fibroblasts (gift from Prof. Nancy Braverman) were maintained in DMEM supplemented with 10% fetal bovine serum (FBS), non-essential amino acids, 1% penicillin/streptomycin, at 37 °C/5% CO2. Cells modified via CRISPR/Cas9 were maintained as above with the addition of puromycin (1 µg/ml, Sigma) during the selection of the clonal populations. iPSCs, and patient-derived iPSCs (the Genome Engineering & Stem Cell Center (GESC) at Washington University in St. Louis, Baylor College of Medicine, and Rarebase, PBC) were grown in mTSER1 or mTESRPlus media (STEM CELL Technologies). The concentration of cells for plating was

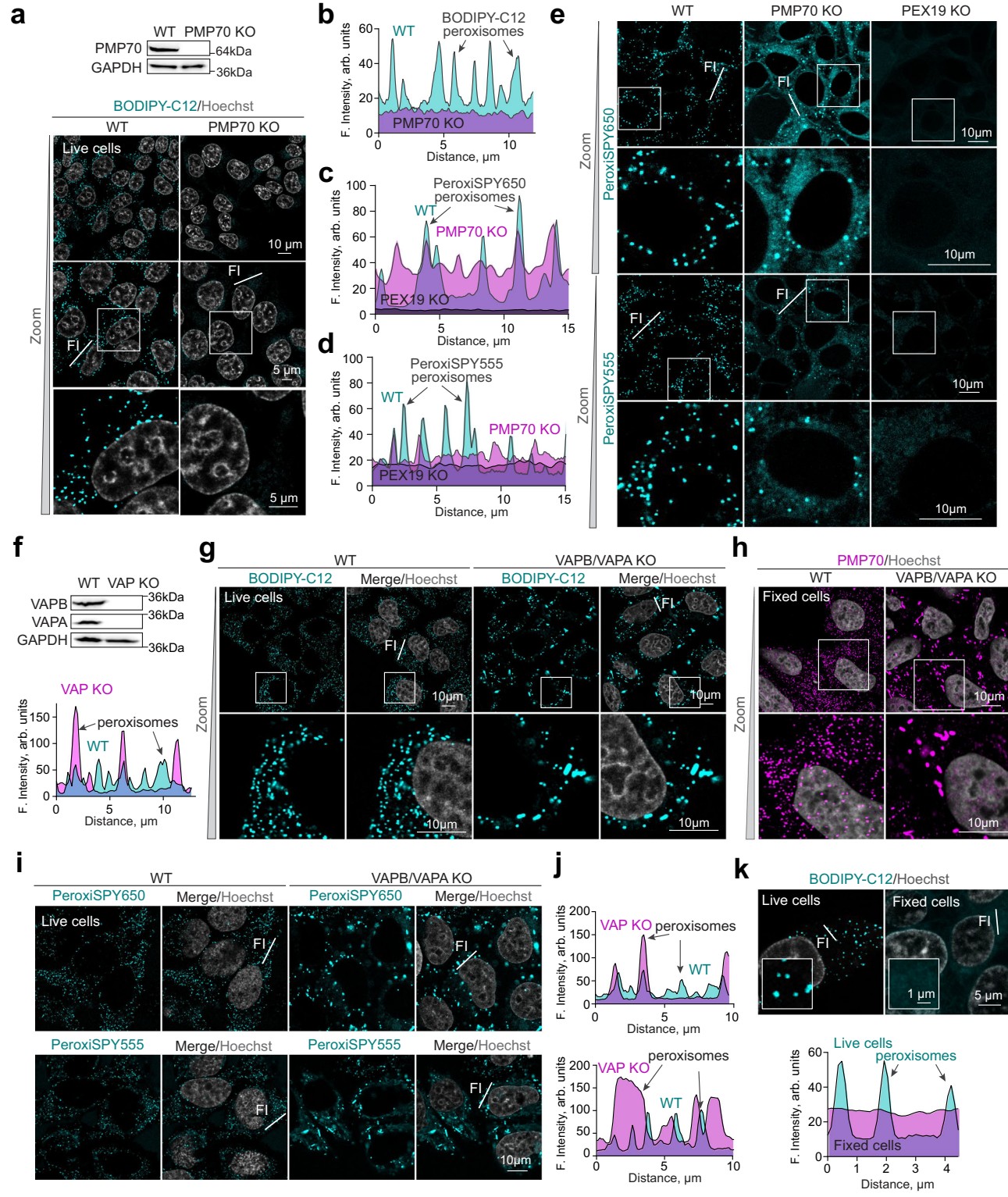

determined using a cell counter (Countess II FL, Life Technologies) with the cell counting chambers (Invitrogen).

## Antibodies

We used the following antibodies: anti-GAPDH (sc-47724, Santa Cruz Biotechnology), anti-PMP70 (SAB4200181, Sigma), anti-PEX19 (14713-1-AP, Proteintech), anti-PEX5 (HPA039260, Sigma), anti-VAPB (14477-1-AP, ProteinTech), anti-PEX10 (STJ119242, St John's Laboratory), anti-VAPA (sc-293278, SantaCruz Biotech) anti-PEX14 (10594-1-AP, Proteintech). Primary antibodies were used at 1:250 dilution.

Secondary antibodies for immunofluorescence: anti-rabbit IgG Cy3-conjugated (Sigma-Aldrich C2306), anti-Mouse IgG Cy3-conjugated (Sigma-Aldrich C2181), anti-rabbit IgG Cy5 conjugated (Invitrogen A10523), Anti-Mouse IgG H&L (Alexa Fluor® 488) (Abcam). Secondary antibodies were used at 1:1000 dilution.

## Chemicals

Hoechst (Sigma), fatty acid-free BSA (PAN), Phenylmethylsulfonyl fluoride (PMSF, Sigma), BODIPY-C12 (BODIPY™ 558/568 C12, D3835, ThermoFisherScientific), BODIPY-FL-C12 (green, D3822,

**Fig. 4 | Peroxisome staining depends on peroxisomal membrane transporter PMP70 (ABCD3). a, b** Confocal microscopy of live WT and PMP70 KO HEK293T cells stained with BODIPY-C12 (0.1 μg/ml) for 10 min. Nuclei were stained with Hoechst (10 μg/ml). Representative overlayed intensity profiles for different cell lines are shown (PMP70 KO(magenta), WT (cyan), PEX19 KO (black)), scale bar − 10 and 5 μm. Western blot shows PMP70 levels in WT and PMP70 KO cells. **c−e** Confocal microscopy of live WT and PMP70 KO HEK293T cells stained with PeroxiSPY650 and PeroxiSPY555 (1 μM) for 10 min. Nuclei were stained with Hoechst (10 μg/ml). Representative overlayed intensity profiles for different cell lines are shown (PMP70 KO(magenta), WT (cyan), PEX19 KO (black)), scale bar − 10 μm. **f, g** Confocal microscopy of live WT and VAPA/VAPB KO HeLa cells stained with BODIPY-C12 (0.1 μg/ml) for 15 min, scale bar − 10 μm. Nuclei were stained with Hoechst (10 μg/ml). **f** Western blot shows VAPB and VAPA levels in WT and KO cells.

Representative overlayed intensity profiles for different cell lines are shown (VAPA/B KO (magenta) and WT (cyan)). **h** Confocal microscopy of fixed WT and VAPA/VAPB KO HeLa cells stained with PMP70 antibody, scale bar − 10 μm. Nuclei were stained with Hoechst (10 μg/ml). **i, j** Confocal microscopy of live WT and VAPA/VAPB KO HeLa cells stained with PeroxiSPY555 and PeroxiSPY650 (1 μM) for 15 min, scale bar − 10 μm. Nuclei were stained with Hoechst (10 μg/ml). Representative intensity profile from separate representative cytoplasmic regions in WT and VAPA/B KO is shown ((VAPA/B KO (magenta) and WT (cyan)). **(k)** Confocal microscopy of live and fixed WT HEK293T cells stained with BODIPY-C12 (1 μg/ml) for 10 min. Nuclei were stained with Hoechst (10 μg/ml), scale bar − 1 and 5 μm. Representative overlayed intensity profiles for different conditions are shown (live cells (cyan) and fixed cells (magenta)).

---

ThermoFisherScientific), Nile Red (Sigma), BODIPY (D3922, Thermo-FisherScientific), BODIPY-Cholesterol (7356, Setareh Biotech), BODIPY-methyl-ester (CellTrace, C34556, ThermoFisherScientific), Capric acid (8797.1, Carl Roth), Glycerin-tripalmitoleate (T2630-100MG, Sigma), Lecithin (9812.1, Carl Roth).

## CRISPR/Cas9

Knockout cell lines were constructed using CRISPR/Cas9 protocol and plasmids described in Ran et al. (Ran et al., 2013). Knockout cell lines were verified by western blotting, immunofluorescence. Genomic DNA was sequenced to verify disrupted region in knockout or the fidelity of endogenous tagging. CRISPR specificity was profiled using Digenome-Seq web tool (http://www.rgenome.net/cas-offinder/) (Bae et al., 2014). Off-targets were not found. The following target sequences are used to modify genomic DNA: knockout of PMP70 on the C-terminus − 5′-CGGCTCGCTGGTACCGGCAGTGCCA-3′, knockout of PEX5 − 5′-GGGGTGCCAACCCGCTCATGAAGCTCGC-3′, knockout of PEX19 − 5′-GGAGGTAGCAAGATGGCCGCCGCTG-3′ (PEX5 and PEX19 are described previously).

## Plasmid construction

All plasmids were constructed using *Escherichia coli* strain DH5α. We used px459 plasmid to clone CRISPR/Cas9 constructs for gene knockouts. pSpCas9(BB)−2A-Puro (PX459) V2.0 was a gift from Feng Zhang (Addgene plasmid # 62988; http://n2t.net/addgene:62988; RRID:Addgene_62988) (Ran et al., 2013). Human PEX5L was amplified from Lentiviral plasmid collection, PEX5L was inserted into Nhe1 EcoR1 sites in pcDNA3.1 with a C-terminal His and Myc tags. We constructed the following plasmids: pcDNA3.1 GFP-SKL, pcDNA3.1 GFP, Px459-PMP70-KO, pcDNA3.1 PEX5L-hismyc.

## Immunofluorescence

Cells were grown on glass bottom plates or glass slides, fixed using 4% paraformaldehyde in phosphate-buffered saline (PBS) for 10 min, washed with PBS, permeabilized with 0.5% Triton X100, washed with PBS, then blocked overnight in 5% BSA in PBS before antibody staining.

## Radioactive measurement of beta fatty acid oxidation using 13,14-3H docosanoic acid (C22:0)

Cells were grown in 12-well plates with initial plating density of 100,000 cells per well for two days. Cold C22:0 (4 μM) and 13,14-3H C22:0 (1 μCi, Anawa) were mixed with the media and added to the cells for 20 h. Media was collected and processed according to[50,51] with minor modifications. Radioactivity was quantified in the media and water fractions transferred to the Scintillation vials after 3 days using scintillation counter. Cold samples, and cell-free samples were used as controls and background subtraction.

## Zebrafish maintenance

Wild-type (AB strain) fish were raised, maintained, and handled according to established protocols[52]. All fish were obtained and kept at

the EPFL Zebrafish facility. All the experiments were conducted using embryos derived from freely mating adults (strain AB, 4−8 months old) and are therefore covered under the general animal experiment license of the EPFL granted by the Service de la Consommation et des Affaires Vétérinaires of the canton of Vaud−Switzerland (authorization number, VD-H23). Zebrafish embryos and larvae were used before they reached a protected developmental stage and do not require the authorization of the Ethics Committee. Zebrafish embryos were raised and maintained in E3 medium at 28.5 °C without light cycle as described in (Westerfield[52]), until the desired developmental stages. Embryo staging was done according to morphological characteristics corresponding to hours post-fertilization or days post-fertilization as described in Kimmel et al.[53].

## Microinjection of the Zebrafish embryos

mRNAs were in vitro transcribed from T7-GFP-polyA or T7-GFP-SKL-polyA PCR products with mMESSAGE mMACHINE™ T7 Transcription Kit#AM1344 according to the manufacturer instructions and purified by RNeasy Mini Kit (Qiagen 74104). Embryos were collected within 30 min following fertilization and incubated in petri dishes with E3 media. Fertilized eggs were microinjected with 200 ng/mkl of the mRNAs into the single blastomere at the one-cell stage. After injection, the embryos were kept in E3 media at 28.5 °C.

## Peroxisome labeling

To label peroxisomes in live animal cells, grow cells to 40−80% confluency (days 1 or 2 after splitting) on a glass-bottom plate, mix cell culture media with the dye at 1 μM concentration in a separate tube and add back to the cells (fresh culture media can also be used, in this case replace the media with a pre-mixed dye (1 μM)in the fresh cell culture media). Incubate cells for 10 min before imaging.

## Microscopy

For live cell imaging 4-well microscope glass bottom plates (IBIDI) or Cell-view cell culture dish (Greiner Bio One) were used. Alternatively, cells were grown on glass slides (Marienfeld). Confocal images and movies were acquired using a SP8 (Leica) confocal microscope equipped with a temperature and CO2 incubator, using a 60x PlanApo VC oil objective NA 1.40. We used 405 nm laser excitation (420−460 emission band) for Hoechst, 488 nm laser excitation (500-540 emission band) for BODIPY-FL-C12 (green) dye and Bodipy-Cholesterol, 561 nm laser excitation (575−615 emission band) for BODIPY-C12, PeroxiSPY555, BODIPY-methyl-ester, and MaP-C18 dyes, and 640 nm laser excitation (655−710 emission band) for PeroxiSPY650, SiR-C16, and SiR-C18 dyes. Image processing was performed using Fiji (ImageJ) software.

## Zebrafish microscopy

At selected developmental stages, embryos were manually dechorionated, incubated with BODIPY-C12 or PeroxiSPY650 for indicated amounts of time in the fish water, and oriented by embedding them in

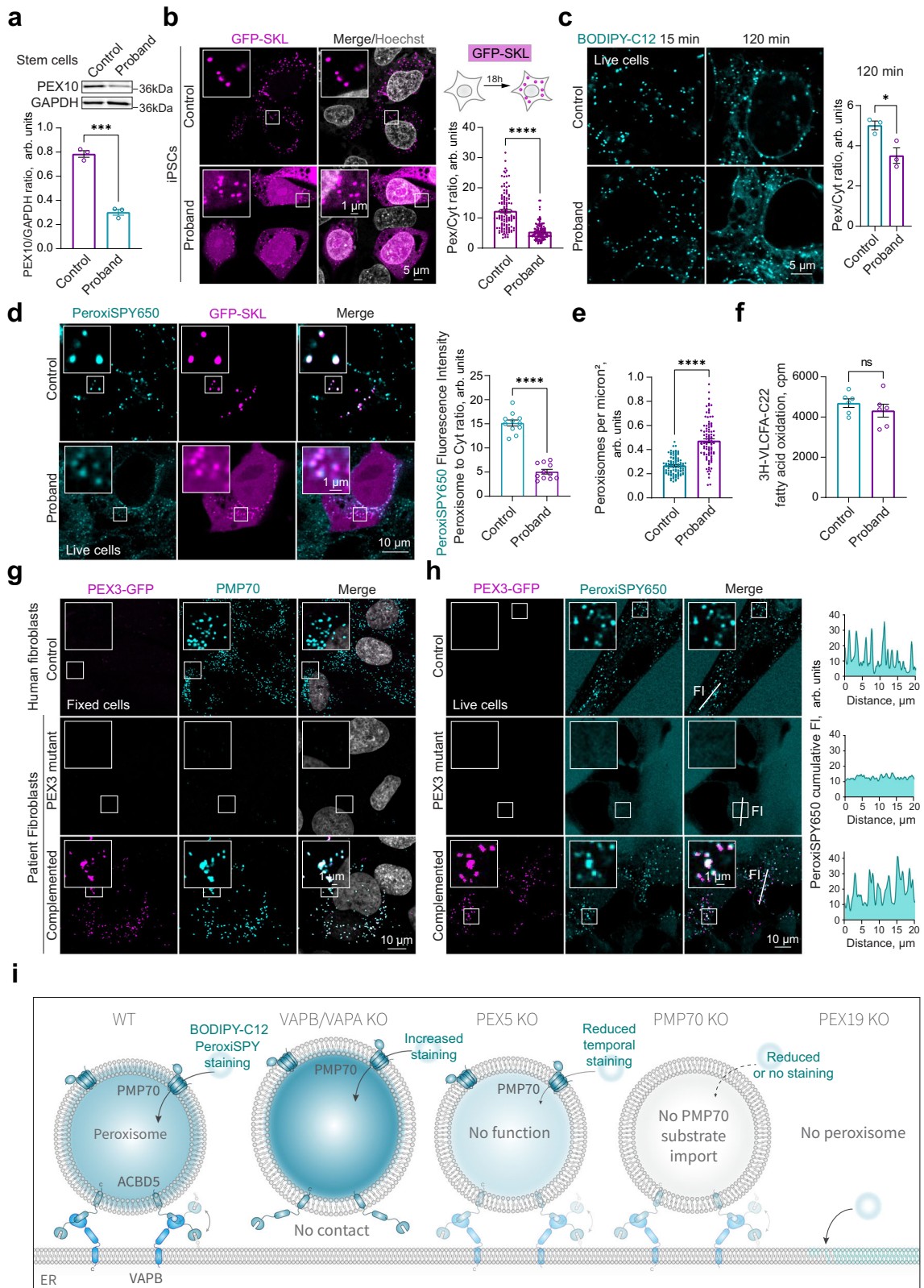

i

a drop of 1% agar pads made of the same fish water as the experimental liquid media, covered with a glass slide.

**Plant growth and staining**

35 S:GFP-PTS1(SKL) *A. thaliana* peroxisome marker line seeds[46] were sterilized with 70% ethanol and stratified for 2 days at 4 °C in the dark, then germinated vertically at 22 °C under continuous light (100 μE) on

agar plates containing half-strength Murashige and Skoog (MS) medium[54].

For seedling staining, 21-day-old *A. thaliana* GFP-SKL seedlings were transferred from plates into (1 μM) dye solutions diluted in half-strength liquid MS media. After incubation, roots were mounted on slides and imaged. Protoplast isolation from 3-week-old *A. thaliana* leaves was performed using "TAPE" sandwich method[55]. Isolated

**Fig. 5 | Peroxisome staining enables identifying peroxisome abnormalities in patient cells. a** Western blot of PEX10 in control and proband patient-derived iPSCs. Quantification shows the ratio of PEX10 to GAPDH in control (magenta) and Proband (cyan), mean ± SEM, *** - $p < 0.001$, $N = 3$ biological repeats, two-tailed unpaired $T$ test. **b** Confocal microscopy of fixed control and proband patient-derived iPSCs expressing GFP-SKL. Nuclei were stained with Hoechst (10 μg/ml), scale bar – 5μm. Quantification shows the ratio of peroxisome to the cytoplasm fluorescence intensity of GFP-SKL marker in control (cyan) and Proband (magenta), mean ± SEM, **** - $p < 0.0001$, $N = 120$ cells, two-tailed Mann-Whitney. **c** Confocal microscopy of live control and proband patient-derived iPSCs stained with BODIPY-C12 (0.1 μg/ml) for indicated amounts of time. Nuclei were stained with Hoechst (10 μg/ml), scale bar – 5 μm. Quantification shows the ratio of peroxisome to cytoplasm fluorescence intensity of BODIPY-C12 in control (cyan) and Proband (magenta), mean ± SEM, * - $p = 0.0267$, $N = 3$ replicas of a ratio of peroxisome to cytoplasm fluorescence intensity pooled from > 300 peroxisomes per replica, two-tailed unpaired $T$ test. **d, e** Confocal microscopy of live control and proband patient-derived iPSCs expressing GFP-SKL and stained with PeroxiSPY650 (1 μM),

scale bar – 1 and 10 μm. Quantification shows the ratio of peroxisome to cytoplasm fluorescence intensity of PeroxiSPY650 in control (cyan) and Proband (magenta), mean ± SEM, *** - $p < 0.001$, $N = 9$ replicas pooled from 300 peroxisomes per sample per repeat. **e** Quantification shows the number of peroxisomes per square micron, mean ± SEM, **** - $p < 0.0001$, $N = 100$ cells, two-tailed unpaired $T$ test. **f** Radioactive 3H-C22 fatty acid oxidation in control and Proband iPSCs. Quantification shows counts per minute (cpm) after 24 h of oxidation in control (cyan) and Proband (magenta), mean ± SEM, $N = 6$ replicas, two-tailed unpaired $T$ test. **g** Confocal microscopy of fixed control, PEX3 null patient-derived fibroblasts, and patient fibroblasts complemented with PEX3-GFP. Peroxisomes were stained with PMP70 antibody, nuclei were stained with Hoechst (10 μg/ml), scale bar – 10 μm. **h** Confocal microscopy of live control, PEX3 null patient-derived fibroblasts, and patient fibroblasts complemented with PEX3-GFP, stained with PeroxiSPY650 (1 μM) for 15 min. Representative cumulative intensity profiles are shown. **i** Schematic of BODIPY-C12 (B-C12) staining of peroxisomes in WT, VAP KO, PMP70 KO, PEX5 KO, and PEX19 KO cells.

protoplasts were resuspended in dye-containing solutions diluted in W5 buffer (154 mM NaCl, 125 mM CaCl2, 5 mM KCl, 5 mM glucose, 1.5 mM MES, pH 5.6). After incubation with (1 μM) dye solution, protoplast suspensions were transferred to microscope slides for imaging.

## Toxicity assay

LDH release was quantified according to the manufacturer instruction (Cytotoxicity Detection KitPLUS (LDH), 4744926001, Sigma).

Cell growth assay: HeLa cells were seeded at a low density (3000 cells/well) on 96-well plates (Phenoplate-96, PerkinElmer), cultured overnight, and treated with the probes at the fixed range of concentrations from 5120 to 5 nM, using eleven steps of two-fold serial dilutions. Three replicates were performed for each condition. The cells were grown 48 h in the presence of the probes. At 48 h, Hoechst33342 was added to each well (final concentration 5 g/mL) to count cell nuclei by Hoechst fluorescence. Images were acquired with a GE INCell2200 automated fluorescence microscope, using a 4X/0.2 objective, and 4 field-of-views were recorded per well to cover the entire well surface.

Cell counting was performed using a custom image analysis pipeline in CellProfiler (v4.2.5). Images were corrected for illumination and rescaled, and nuclei were segmented using the global minimum cross-entropy thresholding method. Total cell counts per well were then plotted versus probe concentration.

## Chemical synthesis

All chemical reagents and anhydrous solvents for synthesis were purchased from commercial suppliers (Sigma-Aldrich, Fluka, Acros) and were used without further purification or distillation. 19-eicosynoic acid was custom synthesized by Spirochrome AG. The composition of mixed solvents is given by the volume ratio (v/v). 1H nuclear magnetic resonance (NMR) spectra were recorded on a Bruker DPX 400 (400 MHz for 1H) with chemical shifts (δ) reported in ppm relative to the solvent residual signals (7.26 ppm for of CDCl3; 3.31 ppm for MeOD). Coupling constants are reported in Hz. LC-MS was performed on a Shimadzu MS2020 connected to a Nexerra UHPLC system equipped with a Waters ACQUITY UPLC BEH Phenyl 1.7 μm 2.1 x 50mm column. Buffer A: 0.05% HCOOH in H2O Buffer B: 0.05% HCOOH in acetonitrile. LC gradient: 10% to 90% B within 6.0 min with 0.5 ml/min flow. Unless otherwise stated, preparative HPLC was performed on a Dionex system equipped with an UltiMate 3000 diode array detector for product visualization on a Waters SymmetryPrep C18 column (7 μm, 7.8 × 300 mm). Buffer A: 0.1% v/v TFA in H2O; Buffer B: acetonitrile. Gradient was from 25% to 90% B within 30 min with 3 ml/min flow.

**SiR-C16.** SiR-azide (Spirochrome # SC009) (5.0 mg, 9.0 umol, 1 eq.) was dissolved in DMSO (400 ul). Separately, 15-hexadecynoic acid (2.5 mg, 9.9 μmol, 1.1 eq.) was dissolved in 400 ul dioxane. The two solutions were mixed. In a separate vial, TBTA (9 μL of 0.1 M DMSO solution, 0.9 μmol, 0.1 eq.) and CuSO4 (9 μL of 0.1 M aqueous solution, 0.9μmol, 0.1 eq.) were mixed and immediately added to the reaction mixture. Finally, sodium ascorbate (9 μl of 0.5 M aqueous solution, 4.5μmol, 0.5 eq.) was added to the reaction. After 2 h the product was purified by preparative HPLC. The fractions containing the product were pooled and evaporated under reduced pressure. Yield 7.0 mg (84%) as TFA salt.

HRMS: [M]+ Calcd for C46H63N6O5Si + 807.4624; Found 807.4636. 1H NMR (400 MHz, MeOD) δ 8.07 – 7.97 (m, 2H), 7.72 (s, 1H), 7.67 (s, 1H), 7.07 (d, J = 2.9 Hz, 2H), 6.73 (d, J = 9.0 Hz, 2H), 6.64 (dd, J = 9.0, 2.9 Hz, 2H), 4.40 (t, J = 6.8 Hz, 2H), 3.37 (t, J = 6.7 Hz, 2H), 2.99 (s, 12H), 2.61 (t, J = 7.6 Hz, 2H), 2.26 (t, J = 7.4 Hz, 2H), 2.17 (p, J = 6.7 Hz, 2H), 1.64 – 1.51 (m, 4H), 1.37 – 1.21 (m, 18H), 0.65 (s, 3H), 0.55 (s, 3H).

**SIR-C18.** 17-Octadecynoic Acid (2.5 mg, 9 mmol, 1 eq) was mixed with SIR-azide (5 mg, 9 mmol, 1 eq) in 900 mL dry DMSO. TBTA (4.8 mg, 9 mmol, 1 eq) was dissolved in 80 mL dry DMSO and added to a mixture of CuSO4 (1.4 mg, 9 mmol, 1 eq) dissolved in 10 mL H2O. The mixture of TBTA and CuSO4 was then combined with the mixture of the 17-Octadecynoic Acid and SIR-azide, and the reaction started by the addition of sodium ascorbate (26.7 mg, 135 mmol, 15 eq), dissolved in 269.5 mL H2O. The reaction mixture was stirred for 1 h at room temperature and then the solids are pelleted by centrifugation. The supernatant was subjected to FC (4% MEOH in DCM), followed by HPLC using a VP 250/21 Nucleodur C18 Pyramid column and a gradient of A (H2O with 0.1% TFA) and B (MeCN): 0–3 min, 100% A, 3–23 min 100·0% A, 23–27 min, 0–100% A. The retention time of the dark blue product is 24.2 min. Yield: 3.4 mg, 4.07 mmol, 45.2%. HRMS: [M + H]+ calculated: 835.4937, found 835.4941. 1H NMR (400 MHz, CDCl3) δ 8.05 (d, J = 8.0 Hz, 1H), 7.96 (d, J = 8.2 Hz, 1H), 7.74 (s, 1H), 7.50 (s, 1H), 7.42 (s, 1H), 7.25 (d, J = 2.9 Hz, 2H), 6.94 (d, J = 9.0 Hz, 2H), 6.81 (dd, J = 9.1, 2.8 Hz, 2H), 4.43 (t, J = 6.5 Hz, 2H), 3.47 – 3.39 (m, 2H), 3.11 (s, 13H), 2.69 (t, J = 7.7 Hz, 2H), 2.30 (t, J = 7.4 Hz, 2H), 2.24 – 2.16 (m, 2H), 1.66 – 1.54 (m, 4H), 1.37 – 1.19 (m, 22H), 0.68 (s, 3H), 0.59 (s, 3H).

**Peroxy-SPY650.** SiR-azide (Spirochrome # SC009) (5.0 mg, 9.0 μmol, 1 eq.) was dissolved in DMSO (400 μl). Separately, 19-eicosynoic acid (2.8 mg, 9.9 μmol, 1.1 eq.) was dissolved in 400 ul dioxane. The two solutions were mixed. In a separate vial, TBTA (9 μL of 0.1 M DMSO solution, 0.9 μmol, 0.1 eq.) and CuSO4 (9 μL of 0.1 M aqueous solution, 0.9 μmol, 0.1 eq.) were mixed and immediately added to the reaction mixture. Finally, sodium ascorbate (9 μl of 0.5 M aqueous solution,

4.5 µmol, 0.5 eq.) was added to the reaction. After 2 h, the product was purified by preparative HPLC. The fractions containing the product were pooled and evaporated under reduced pressure. Yield 8.2 mg (87%) as TFA salt. HRMS m/z: [M]+ Calcd for C50H71N6O5Si + 863.5250; Found 863.5248. 1H NMR (400 MHz, MeOD) δ 8.76 (t, J = 5.7 Hz, 1H), 8.28 (d, J = 8.2 Hz, 1H), 8.09 (dd, J = 8.2, 1.8 Hz, 1H), 7.77 (s, 1H), 7.69 (d, J = 1.7 Hz, 1H), 7.31 (d, J = 2.8 Hz, 2H), 6.95 (d, J = 9.5 Hz, 2H), 6.75 (dd, J = 9.5, 2.9 Hz, 2H), 4.45 (t, J = 6.8 Hz, 2H), 3.46 – 3.38 (m, 2H), 3.28 (s, 13H), 2.64 (t, J = 7.6 Hz, 2H), 2.31 – 2.15 (m, 4H), 1.67 – 1.53 (m, 5H), 1.37 – 1.24 (m, 28H), 0.64 (s, 3H), 0.59 (s, 3H).

**MaP555-azide.** Map555-COOH[30] (10 mg, 18.6 µmol, 1 eq.) was dissolved in DMSO (0.5 ml). Diisopropylamine (11.2 ul, 65.1 µmol, 3.5 eq.) and TSTU (6.7 mg, 22.3 µmol, 1.2 eq.) were successively added. After 5 min, 3-azido-1-propanamine (2.2 µl, 22.3 µmol, 1.2 eq.) was added. After 30 min, the product was purified by preparative HPLC. The fractions containing the product were pooled and lyophilized. Yield 10.7 mg (78%) as TFA salt. HRMS m/z: [M]+ Calcd for C30H35N8O5S + 619.2446; Found 619.2452. 1H NMR (400 MHz, MeOD) δ 8.77 (br, s, 1H), 8.17 (d, J = 8.1 Hz, 1H), 8.02 (d, J = 8.1 Hz, 1H), 7.78 (br, s, 1H), 7.20 – 6.71 (m, 6H), 3.50 – 3.42 (m, 2H), 3.38 (t, J = 6.6 Hz, 2H), 3.22 (br, s, 12H), 2.62 (s, 6H), 1.85 (p, J = 6.7 Hz, 2H).

**MaP555-C18.** MaP555-azide TFA salt (3.0 mg, 3.0 µmol, 1 eq.) was dissolved in DMSO (250 ul). Separately, 17-octadecynoic acid (0.9 mg, 3.3 µmol, 1.1 eq.) was dissolved in 250 µl dioxane. The two solutions were mixed. In a separate vial, TBTA (3 µL of 0.1 M DMSO solution, 0.3 µmol, 0.1 eq.) and CuSO4 (3 µL of 0.1 M aqueous solution, 0.9 µmol, 0.1 eq.) were mixed and immediately added to the reaction mixture. Finally, sodium ascorbate (3 µl of 0.5 M aqueous solution, 1.5 µmol, 0.5 eq.) was added to the reaction. After 2 h the product was purified by preparative HPLC. The fractions containing the product were pooled and evaporated under reduced pressure. Yield 2.1 mg (69%) as TFA salt. HRMS m/z: [M]+ Calcd for C48H67N8O7S + 899.4848; Found 899.4850. 1H NMR (400 MHz, MeOD) δ 8.17 (d, J = 8.1 Hz, 1H), 8.03 (d, J = 8.1 Hz, 1H), 7.90 – 7.71 (m, 2H), 7.26 – 6.85 (br, m, 6H), 4.45 (t, J = 6.8 Hz, 2H), 3.43 (t, J = 6.3 Hz, 2H), 3.28 (s, 12H), 2.70 – 2.57 (m, 8H), 2.31 – 2.15 (m, 4H), 1.60 (dt, J = 14.4, 7.3 Hz, 2H), 1.40 – 1.21 (m, 24H).

**PeroxySPY555.** MaP555-azide TFA salt (3.0 mg, 3.0 µmol, 1 eq.) was dissolved in DMSO (250 µl). Separately, 19-eicosynoic acid (1.0 mg, 3.3 µmol, 1.1 eq.) was dissolved in 250 ul dioxane. The two solutions were mixed. In a separate vial, TBTA (3 µL of 0.1 M DMSO solution, 0.3 µmol, 0.1 eq.) and CuSO4 (3 µL of 0.1 M aqueous solution, 0.9 µmol, 0.1 eq.) were mixed and immediately added to the reaction mixture. Finally, sodium ascorbate (3 µl of 0.5 M aqueous solution, 1.5 µmol, 0.5 eq.) was added to the reaction. After 2 h the product was purified by preparative HPLC. The fractions containing the product were pooled and evaporated under reduced pressure. Yield 2.2 mg (70%) as TFA salt. HRMS m/z: [M]+ Calcd for C50H71N8O7S + 927.5161; Found 927.5161. 1H NMR (400 MHz, MeOD) δ 8.13 (d, J = 8.2 Hz, 1H), 8.02 (d, J = 8.1 Hz, 1H), 7.82 – 7.65 (m, 1H), 7.38 – 6.40 (m, 6H), 4.42 (t, J = 6.8 Hz, 2H), 3.40 (t, J = 6.7 Hz, 2H), 3.20 (s, 12H), 2.67 – 2.58 (m, 8H), 2.27 (t, J = 7.4 Hz, 2H), 2.23 – 2.13 (m, 2H), 1.66 – 1.53 (m, 3H), 1.33 – 1.24 (m, 28H).

### Statistics and reproducibility
Three or more independent experiments were performed to obtain the data. Each experiment was independently repeated three times with similar results. P values were calculated by two-tailed Student t-test, or one-way ANOVA for samples following normal distribution determined by the Shapiro-Wilks test. The equality of variances was verified by Brown-Forsythe or F test. Mann-Whitney (2 groups), or Kruskal-Wallis (multiple groups) tests were used for samples that didn't follow a normal distribution. No statistical method was used to predetermine sample size. No data were excluded from the analyses; the investigators were not blinded to allocation during experiments and outcome assessment. Microscopy is performed from a randomized regions to avoid ROI bias. Repeats and replicas refer to biological repeats. Refer to the Source Data file for the details on statistical analysis.

### Reporting summary
Further information on research design is available in the Nature Portfolio Reporting Summary linked to this article.

## Data availability
All data needed to evaluate the conclusions of the paper are present in the paper and supplementary materials. The data generated in this study are provided in the Supplementary Information/Source data file. All the reagents generated in this paper are available upon request from the corresponding author, TA. PeroxiSPY555 and PeroxiSPY650 will be commercially available at Spirochrome AG. Source data are provided with this paper.

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

## Acknowledgements

We thank Prof. Lindy Holden-Dye, Prof. Gisou van der Goot, and Prof. Giovanni D'Angelo for their support, including access to essential

equipment, reagents, and discussions. We thank the EPFL Gene Expression Core Facility (GECF) for providing access to essential equipment and lenti-viral collection, the EPFL Bioimaging and Optics Core facility (PTBIOP) for access to essential imaging equipment, Florian Lang, Paula Barrera Gomez and the EPFL Center of Phenogenomics (CPG) zebrafish facility for providing access to essential equipment and fish maintenance, Biomolecular Screening Core Facility (BSCF) for providing the SiR dye. We thank Pietro De Camilli for the gift of VAPA/VAPB KO cells. We thank Prof. Ronald J. A. Wanders for the advice on radioactive measurements of peroxisomal fatty acid oxidation. We thank Prof. Nancy Elise Braverman, Prof. Heidi Mc Bride, and Aurèle Besse-Patin for a gift of PEX3-deffient patient fibroblasts. We thank Prof. Daniel Kaganovich for the comments on the manuscript and a gift of essential reagents. We thank Andrew Longenecker and PBD Project, Inc, a non-profit focused on finding treatments for Peroxisome Biogenesis Disorder patients, for the generous gift of the PBD patient cell lines that were created in collaboration with The Genome Engineering & Stem Cell Center (GESC) at Washington University in St. Louis, Baylor College of Medicine, and Rarebase, PBC. We thank Prof. Bonnie Bartel from Rice University (Texas, USA) for the gift of the *A. thaliana* transgenic PTS1(SKL)-GFP peroxisomal marker line. TA was funded by the HFSP Long-term Fellowship (LT000559/2021-L) and supported by the EPFL Faculty and the University of Southampton. MS is supported by the ELISIR program of the EPFL School of Life Sciences. AB was funded by the HFSPO Scientists for Scientists Initiative to help scientists affected by the war in Ukraine.

## Author contributions

D.K., A.B., A.G., M.B., F.K., L.R., M.S., and T.A. performed the experiments. D.K. and T.A. performed Zebrafish experiments; M.S. and L.R. performed and designed the chemical synthesis; A.G. and T.A. performed the plant experiments; L.R., M.B., F.K., and T.A. performed the toxicity tests; T.A. designed the study, performed the data analysis, wrote the manuscript, and supervised the study.

## Competing interests

L.R. owns shares of Spirochrome A.G., commercializing PeroxiSPY probes. The remaining authors declare no competing interests.
