## [Peer Review File · Nature Communications]

Reviewers' Comments:

Reviewer #1:

Remarks to the Author:

Peroxisomes are membrane-bounded organelles essential for various aspects of lipid metabolism. Specific staining of peroxisomes is critical for imaging peroxisome dynamics in live cells and organisms. Currently, visualization of peroxisomes in live cells is only possible using fluorescent-protein or peptide fusions to the Ser-Lys-Leu (SKL) peroxisome targeting signal. Such fusion constructs must either be genetically encoded or directly transfected into cells, which is time-consuming and may not be applicable for sensitive cell types or whole-tissue samples.

This study reports the unexpected discovery that peroxisomes can be quickly and reliably labeled in live cells using the commercially available fluorescent fatty-acid analog BODIPY-C12. While this probe has traditionally been used to label lipid droplets, the study shows that it also prominently accumulates in peroxisomes. Peroxisomal staining by BODIPY-C12 is demonstrated in several different human cell lines, including patient-derived iPSCs, and in live zebrafish embryos. The staining requires the peroxisomal fatty-acid transporter PMP70 and is independent of protein import. Nonetheless, staining is less stable in cells with peroxisomal protein import defects, which may help distinguish functional from dysfunctional peroxisomes in disease settings. The authors further develop a novel fluorescent fatty-acid analog called SiR-C18 with improved peroxisome specificity, and claim that it can identify morphological differences of peroxisomes in cells from a patient with a mild peroxisome-biogenesis mutation.

Overall, the study convincingly describes a new method for labeling peroxisomes in live cells and tissues using lipid-based probes. The development of a peroxisome-specific probe is impressive; it will undoubtedly be useful for the scientific community and may also have clinical applications. However, the paper deals mostly with BODIPY-C12 whereas the new probe is not extensively characterized. The utility of the new probe to detect peroxisome dysfunction in patient samples is also tenuous. While arguably not the main focus of the paper, the mechanism by which lipid-based probes stain peroxisomes and are incorporated into cellular lipids remains speculative.

I recommend the paper to be eligible for publication if the following specific points are addressed.

Specific points:

Co-localization of SiR-C18 with a peroxisomal marker (e.g., GFP-SKL) should be shown in tissue-culture cells as well as in zebrafish embryos.

The stability and kinetics of peroxisomal staining by SiR-C18 should be assessed. Specifically, staining by SiR-C18 should be evaluated after different incubation times and at different concentrations of the probe, as done for BODIPY-C12.

The authors should test the dependence of peroxisomal staining by SiR-C18 on peroxisomal protein import and on PMP70-mediated fatty-acid import. Specifically, peroxisomal staining by SiR-C18 should be compared between PEX5 KO, PMP70 KO, and VAPA/B KO cells.

Cytotoxicity of SiR-C18 should be evaluated over time at the concentrations used in the study. This information is essential for readers who may want to use the probe for longer-term imaging.

The use of SiR-C18 to identify peroxisomal abnormalities in patient cells should be demonstrated for at least one other disease mutation, as well as in an additional cell type such as fibroblasts.

In Figs. 3D and E, why are the data not included for PEX5 KO and complemented cells stained with low concentrations of BODIPY-C12 (0.1 $\mu\text{g}/\text{ml}$)? It would be important to know whether or not lower concentrations of BODIPY-C12 also stably stain peroxisomes in PEX5 KO cells.

In Fig. 5C, the probe-competition experiment implies that SiR-C18 is preferentially imported into peroxisomes over the shorter BODIPY-C12. To confirm this conclusion, the authors should test whether SiR-C18 retains its peroxisome-to-cytosol ratio at higher concentrations of BODIPY-C12.

The TLC analysis in Figs. 4D and E should be removed because it is incomplete and irrelevant to the story. Peroxisomal staining by BODIPY-C12 is shown to dissipate after 1 h, while no difference in phospholipid incorporation is detectable by TLC after this time. Incorporation of the probe into lipids occurs only after 24 h, which is on a completely different timescale. This interesting observation suggests that BODIPY-C12 could be used to monitor fatty-acid metabolism and could perhaps be later explored by the authors as a possible application of the probe.

Minor concerns:

BODIPY-C12 no longer labels peroxisomes in cells lacking PMP70, which is interpreted to mean that the probe enters peroxisomes through this fatty-acid transporter. However, can the authors disprove the alternative hypotheses, that lack of PMP70 alters some property of peroxisomes (e.g., the lipid content of the peroxisomal membrane) that changes their sensitivity to the probe, or causes mislocalization of some other protein that is actually required for import? Please clarify.

VAPA/B KO cells are used to argue that BODIPY-C12 does not enter peroxisomes via contact sites with the ER. What about contacts with other organelles, notably mitochondria and lipid droplets, which may also accumulate the probe but not necessarily use VAP proteins? Please clarify.

It is unclear why BODIPY-C12 no longer labels peroxisomes after prolonged incubation. If the cells were bathed in media containing the probe and imaged over time, they should continually import the probe. Also, why is it relevant that peroxisomal staining by high concentrations of BODIPY-C12 dissipates over time, when low concentrations stain more stably? Please clarify.

Peroxisomal staining by BODIPY-C12 is stronger in VAPA/B KO cells, suggesting that contact sites with the ER may actually facilitate the export of the probe out of peroxisomes. Does peroxisomal staining dissipate over time in VAPA/B KO cells as it does in WT cells?

In Fig. 5, it would be helpful if BODIPY-C12 and BODIPY-C12 (green) were directly labeled using the names themselves. The use of colored font to distinguish them is confusing. This issue is particularly problematic in panels E and F, where not even colored font is used.

For many of the line scans, it is difficult to tell which region of an image was analyzed to generate the fluorescence profile; perhaps the lines in these images could be thickened. In Figs. 3A, 4B-C, 4F, 5D-F, and figs S1A-D, lines delineating the scanned regions are missing altogether.

For statistical analyses, the authors should specify the units in which their sample sizes (N) are reported. For example, in Fig. 3C, "N=300" presumably means 300 cells; and in Figs. 5B and C, it is unclear what is meant by "N=3 from >1000 pooled peroxisomes" (cells or replicates).

The relevance of fig. S2A is unclear, as it appears to show the same thing as Fig. 2A. Is it supposed to show BODIPY-C12 staining in a different region of the same embryo as in Fig. 2A, or an independently-stained embryo, or an embryo after a different time post-fertilization?

Bodipy is more commonly capitalized as BODIPY.

In Fig. 2A, it would be helpful to outline the region of the embryo magnified in the lower panels .

Line 75: The sentence should read "Nor did lysosome staining...".

Line 87: The sentence should be rewritten as "As a model organism we chose the zebrafish, *Danio rerio*. We stained...".

Line 93: The sentence should read "... we injected single-cell-stage embryos...".

Lines 145-147: A citation of the patient-derived PEX10 mutation and iPS cell line is missing.

Line 148: There should be a period after "... import into peroxisomes." Also, the next sentence

would be simpler if it were written "In PEX10 mutant cells, import of GFP-SKL was delayed after...".

Line 372: In the legend to the quantitation in Fig. 3C, the label should be "(c)" instead of "(d)".

Reviewer #2:

Remarks to the Author:

I have looked at the construction of the paper and the results and I find them noteworthy. The reports on the live-cell imaging of peroxisomes are not very common and authors demonstrate clean results. Peroxisome imaging is achieved by Bodipy-C1 and siR-C18 derivative and the images are of high quality.

The fluorophores with hydrophobic tails might get incorporated in the plasma membrane or gets into the lipid droplets. So it is interesting that these probes gets into peroxisomes as opposed to LDs. The authors do mention its use for LD in the first sentence with a few citations and others (10.1371/journal.pone.0153522; <https://elifesciences.org/articles/66393>; 10.1017/S1431927620016761), but I am unclear what contributed to those specks being observed. What contributes to these changes vis-a-vis fixed vs live cells? Are they motile and do authors observe interaction with other organelle after sometime? After how much time or until what time, these probes localized themselves in peroxisomes. Do they remain so?

I think this is a good study for live-cell imaging of peroxisomes.

Reviewer #3:

Remarks to the Author:

The manuscript of Daria Korotkova et al entitled "A new fluorescent fatty acid conjugate for live cell imaging of peroxisomes" is an interesting work concerning the application of Bodipy-C12 as a fluorescence probe for functional and dysfunctional peroxisomes in live cells.

Due to the important and innovating results, the manuscript could be accepted for publication but only after a major revision concerning some aspects:

- 1- In the methods' section of microscopy, the authors should attribute each laser to the correspondent fluorescent dye and the emission band recorded for each dye (line 286 and 287).
- 2- In figure 1(b), is the intensity profile along a single peroxisome? In order to be a more representative analysis of the dye's intracellular distribution, the intensity profile should be represented along an entire cell.
- 3- The PCC (Pearson's correlation coefficient) should be determined to quantify the colocalization images of the manuscript.
- 4- Is the intensity profile in figure 3 (a) representing the fluorescence pattern of Bodipy-C12 in WT and PEX5 KO HEK293T cells? It's not very clear. This type of plots, normally, are applied to represent the fluorescence intensity of two or more dyes in order to demonstrate the presence/absence of colocalization along the same spatial region (same image). In this case, the authors should plot the intensity profiles of Bodipy-C12 and GFP-SKL per cell line: one plot of the WT cells and other plot of PEX5 KO HEK293T cells.
In figure 4 (f), the intensity profile represents the distribution of the Bodipy-C12 in different spatial regions, what can be concluded from that?
- 5- In figure 5 (a), the authors shouldn't give the same name for structurally different BODIPYs (Bodipy-C12).
- 6- In figure 5 (a), the microscope images of the PEX19 KO HEK293T cells stained with SiR-C18 appear to be wrong: in the SiR-C18 there is no sign of fluorescence, and in the correspondent merged image there is a residual/ background blue fluorescence (right side of the cell's nucleus).

Reviewers' comments:

Reviewer #1 (Remarks to the Author)

Peroxisomes are membrane-bounded organelles essential for various aspects of lipid metabolism. Specific staining of peroxisomes is critical for imaging peroxisome dynamics in live cells and organisms. Currently, visualization of peroxisomes in live cells is only possible using fluorescent-protein or peptide fusions to the Ser-Lys-Leu (SKL) peroxisome targeting signal. Such fusion constructs must either be genetically encoded or directly transfected into cells, which is time-consuming and may not be applicable for sensitive cell types or whole-tissue samples.

This study reports the unexpected discovery that peroxisomes can be quickly and reliably labeled in live cells using the commercially available fluorescent fatty-acid analog BODIPY-C12. While this probe has traditionally been used to label lipid droplets, the study shows that it also prominently accumulates in peroxisomes. Peroxisomal staining by BODIPY-C12 is demonstrated in several different human cell lines, including patient-derived iPS cells, and in live zebrafish embryos. The staining requires the peroxisomal fatty-acid transporter PMP70 and is independent of protein import. Nonetheless, staining is less stable in cells with peroxisomal protein import defects, which may help distinguish functional from dysfunctional peroxisomes in disease settings. The authors further develop a novel fluorescent fatty-acid analog called SiR-C18 with improved peroxisome specificity, and claim that it can identify morphological differences of peroxisomes in cells from a patient with a mild peroxisome-biogenesis mutation.

Overall, the study convincingly describes a new method for labeling peroxisomes in live cells and tissues using lipid-based probes. The development of a peroxisome-specific probe is impressive; it will undoubtedly be useful for the scientific community and may also have clinical applications. However, the paper deals mostly with BODIPY-C12 whereas the new probe is not extensively characterized. The utility of the new probe to detect peroxisome dysfunction in patient samples is also tenuous. While arguably not the main focus of the paper, the mechanism by which lipid-based probes stain peroxisomes and are incorporated into cellular lipids remains speculative.

I recommend the paper to be eligible for publication if the following specific points are addressed.

(Response) We thank this reviewer for their constructive comments and apologize for a delayed response. It was due to my group relocation to the University of Southampton and a delay in the shipment of an additional patient cell line. In the revised version we synthesized a set of peroxisome dyes to find an optimal fatty acid chain length for better peroxisomal signal (we compare C16, C18, and C20) and assay different fluorescent molecules (SiR far red dye, and MaP555 red dye, in addition to BODIPY-C12). We found that SiR-C20 (referred to further as PeroxiSPY650) performs better than SiR-C18, and synthesized Spy555-C20 (referred further as PeroxiSPY555) (Figure S5a-b, and S6 for toxicity). Please see the answers below.

Specific points:

Co-localization of SiR-C18 with a peroxisomal marker (e.g., GFP-SKL) should be shown in tissue-culture cells as well as in zebrafish embryos.

(Response) We replaced SiR-C18 with PeroxiSPY650 (SiR-C20) for all the experiments as it showed a better peroxisomal staining (see Figure S5a-b).

We added the co-localization of the dye with a peroxisomal marker - PeroxiSPY650 with GFP-SKL in Figure 6d, PeroxiSPY650 and PEX3-GFP in Figure 6h, Zebrafish staining co-localization in Figure 5e-f and Figure S7d.

The stability and kinetics of peroxisomal staining by SiR-C18 should be assessed. Specifically, staining by SiR-C18 should be evaluated after different incubation times and at different concentrations of the probe, as done for BODIPY-C12.

(Response) We included a PeroxiSPY650 (far-red) and PeroxiSPY555 (red) kinetics and peroxisome stability experiments in WT, PEX5 KO, and PEX5 KO complemented cells in Figure 3f-h, S5c-e, similarly to what was included for BODIPY-C12 in the first submission. Importantly we also repeated BODIPY-C12 experiments with 1 μ M instead of 1 μ g/ml where we directly compare the intensities. PeroxiSPY dyes displayed similar long-term staining of peroxisomes.

The authors should test the dependence of peroxisomal staining by SiR-C18 on peroxisomal protein import and on PMP70-mediated fatty-acid import. Specifically, peroxisomal staining by SiR-C18 should be compared between PEX5 KO, PMP70 KO, and VAPA/B KO cells.

(Response) We included a PeroxiSPY650 (far-red) and PeroxiSPY555 (red) kinetics and peroxisome stability experiments in WT, PEX5 KO, and PEX5 KO complemented cells in Figure 3f-h, S5c-e. We also included PMP70 KO experiments in Figure 4c-e. Interestingly while the peroxisome staining was reduced with the new dyes, PeroxiSPY still partially stained peroxisomes without PMP70.

We also included the staining data in VAPA/B KO cells in Figure 4i-j. In these experiments, PeroxiSPY dyes performed similarly to BODIPY-C12.

Cytotoxicity of SiR-C18 should be evaluated over time at the concentrations used in the study. This information is essential for readers who may want to use the probe for longer-term imaging.

(Response) We evaluated the cytotoxicity for all the synthesized dyes over 48h (Figure S6b-c) and also included the LDH release experiments in WT and PEX19 KO cells for the 24h incubation with the dyes used throughout the study – BODIPY-C12, PeroxiSPY555, and PeroxiSPY650 (Figure S6a). We included the range of concentrations used in this study as well as higher concentrations (e.g. 10 μ M). Both assays showed no toxicity in WT cells, except for SiR-C18 effect on cell growth at higher concentrations (Figure S6b-c). Interestingly, 24h incubation with high concentrations of PeroxiSPY-C20 was toxic to PEX19 KO cells (Figure S6a).

The use of SiR-C18 to identify peroxisomal abnormalities in patient cells should be demonstrated for at least one other disease mutation, as well as in an additional cell type such as fibroblasts.

(Response) Prof. Nancy Braverman kindly shared patient PEX3 null fibroblasts that lack peroxisomal compartments. In the revised version we demonstrate that PeroxiSPY650 and PeroxiSPY555 stains peroxisomes in WT human fibroblasts and in the patient PEX3 null fibroblasts complemented with human PEX3-GFP fusion (Figure 6g-h). Interestingly BODIPY-C12 dye staining of peroxisomes in fibroblast was undiscernible from the

background cell staining.

In Figs. 3D and E, why are the data not included for PEX5 KO and complemented cells stained with low concentrations of BODIPY-C12 (0.1 $\mu\text{g/ml}$)? It would be important to know whether or not lower concentrations of BODIPY-C12 also stably stain peroxisomes in PEX5 KO cells.

(Response) We now included measurements of staining with different dye concentrations (0.5, 1, and 2 μM) for both WT and PEX5 KO cells for BODIPY-C12 and PeroxiSPY dyes (Figure S5d-e). We didn't include 0.1 μM , because the intensity was too low to detect. In order to compare between the dyes we repeated the relevant experiments using μM instead of $\mu\text{g/ml}$. What we found is BODIPY-C12 long term 1h staining is more sensitive to high concentrations of the dye than PeroxiSPY staining, and that there was no effect on PEX5 KO transient staining.

In Fig. 5C, the probe-competition experiment implies that SiR-C18 is preferentially imported into peroxisomes over the shorter BODIPY-C12. To confirm this conclusion, the authors should test whether SiR-C18 retains its peroxisome-to-cytosol ratio at higher concentrations of BODIPY-C12.

(Response) Thank you for this suggestion! We repeated the dye displacement experiments in Figure 2f-g. We found that when staining with the mix of PeroxiSPY and BODIPY-C12, increase in PeroxiSPY concentration displaces BODIPY-C12 faster than an increase in BODIPY-C12 concentration displaces PeroxiSPY dye (approximately 10 times faster measured by fluorescence intensity in peroxisomes) (see Figure 2f-g).

The TLC analysis in Figs. 4D and E should be removed because it is incomplete and irrelevant to the story. Peroxisomal staining by BODIPY-C12 is shown to dissipate after 1 h, while no difference in phospholipid incorporation is detectable by TLC after this time. Incorporation of the probe into lipids occurs only after 24 h, which is on a completely different timescale. This interesting observation suggests that BODIPY-C12 could be used to monitor fatty-acid metabolism and could perhaps be later explored by the authors as a possible application of the probe.

(Response) We removed this analysis from the manuscript.

Minor concerns:

BODIPY-C12 no longer labels peroxisomes in cells lacking PMP70, which is interpreted to mean that the probe enters peroxisomes through this fatty-acid transporter. However, can the authors disprove the alternative hypotheses, that lack of PMP70 alters some property of peroxisomes (e.g., the lipid content of the peroxisomal membrane) that changes their sensitivity to the probe, or causes mislocalization of some other protein that is actually required for import? Please clarify.

(Response) It is a good point; at this moment we cannot disprove all the alternative hypotheses. Given that PeroxiSPY dyes partially stain PMP70 KO it could be speculated that PeroxiSPY uses other transporters as well as PMP70, while BODIPY-C12 only uses PMP70, or, as reviewer mentioned, the property of the membrane is changed. The only indirect indication of transport into peroxisomes is inability of the dyes to stain fixed cells (membrane

properties changes would be similar in fixed and live cells). We are studying the mechanism further to prove or disprove the direct transport.

VAPA/B KO cells are used to argue that BODIPY-C12 does not enter peroxisomes via contact sites with the ER. What about contacts with other organelles, notably mitochondria and lipid droplets, which may also accumulate the probe but not necessarily use VAP proteins? Please clarify.

(Response) In this manuscript we mainly focused on the dye identification and included only a few potentially relevant experiments to test its specificity (e.g, PEX19 KO, and PEX5 KO). The contact site regulation is interesting as there is a known lipid exchange between peroxisome and other compartments, however examining all the contact sites warrants a new project. At the moment we cannot disprove that dye accumulation happens through other membranes or from other compartments.

It is unclear why BODIPY-C12 no longer labels peroxisomes after prolonged incubation. If the cells were bathed in media containing the probe and imaged over time, they should continually import the probe. Also, why is it relevant that peroxisomal staining by high concentrations of BODIPY-C12 dissipates over time, when low concentrations stain more stably? Please clarify.

(Response) We interpret this as BODIPY-C12 exceeding the capacity (either transporter import or membrane accumulation) of peroxisomes to uptake the dye. Peroxisomes are small compartments and regardless of how they accumulate the dye (membrane or import) a certain amount will exceed this capacity. We can speculate what happens next with the dye by looking at the cells with fewer or dysfunctional peroxisomes (e.g. PEX5 KO) where it accumulates in the cellular membranes. We think that the ability of peroxisomes to degrade or export the dye limits the amount of the dye we use for specific staining. When we normalized concentration to 1 μ M – both BODIPY-C12 and PeroxiSPY exhibited stable staining (Figure 3d-g), adding more dye resulted in a significant increase in the background (Figure 2f) even after just 10-15 min of incubation.

Peroxisomal staining by BODIPY-C12 is stronger in VAPA/B KO cells, suggesting that contact sites with the ER may actually facilitate the export of the probe out of peroxisomes. Does peroxisomal staining dissipate over time in VAPA/B KO cells as it does in WT cells?

(Response) That is a very interesting point. We are currently working on a follow up manuscript on that.

Figure for the reviewer (not part of the manuscript): Confocal microscopy of HeLa WT and VAPA/B KO cells stained with PeroxiSPY650 after 15 min and after 15min followed by washing with fresh media and 30min incubation.

Normally, the staining disappears if we wash out the dye from the media, however VAPA/B KO retain the dye, which explains the increased intensity of staining. As this reviewer also noticed, we think that the dye exits peroxisomes through peroxisome-ER contacts. We are exploring it in the follow up project.

In Fig. 5, it would be helpful if BODIPY-C12 and BODIPY-C12 (green) were directly labeled using the names themselves. The use of colored font to distinguish them is confusing. This issue is particularly problematic in panels E and F, where not even colored font is used.

(Response) We now used BODIPY-FL-C12 label for green version of the dye.

For many of the line scans, it is difficult to tell which region of an image was analyzed to generate the fluorescence profile; perhaps the lines in these images could be thickened. In Figs. 3A, 4B-C, 4F, 5D-F, and figs S1A-D, lines delineating the scanned regions are missing altogether.

(Response) We apologize for that; the intention was not to obscure the image beneath the line. In the revised version we thickened the lines, and added the missing lines to the indicated and new Figures (Figure 3a, 4a-d, 4c is a new Figure 4k; Figure 4f, Figure 5d-f is new 2b-e, and FigureS1A-D).

For statistical analyses, the authors should specify the units in which their sample sizes (N) are reported. For example, in Fig. 3C, "N=300" presumably means 300 cells; and in Figs. 5B and C, it is unclear what is meant by "N=3 from >1000 pooled peroxisomes" (cells or replicates).

(Response) In the revised version we included clarification on the N in the legends for all the figures. In Figure 3c, N=300 cells; Figures 5b and 5c were replaced with a displacement assay (new Figure 2f-g), here we clarified that N=3 replicas of an average peroxisome fluorescence intensity pooled from >300 peroxisomes per replica.

The relevance of fig. S2A is unclear, as it appears to show the same thing as Fig. 2A. Is it supposed to show BODIPY-C12 staining in a different region of the same embryo as in Fig. 2A, or an independently-stained embryo, or an embryo after a different time post-fertilization?

(Response) This is the same embryo, we removed this figure.

Bodipy is more commonly capitalized as BODIPY.

(Response) We changed the name throughout the manuscript.

In Fig. 2A, it would be helpful to outline the region of the embryo magnified in the lower panels.

(Response) We outlined the part of the embryo.

Line 75: The sentence should read "Nor did lysosome staining...".

(Response) Thank you, corrected.

Line 87: The sentence should be rewritten as "As a model organism we chose the zebrafish, *Danio rerio*. We stained...".

(Response) Corrected.

Line 93: The sentence should read "... we injected single-cell-stage embryos...".

(Response) Corrected

Lines 145-147: A citation of the patient-derived PEX10 mutation and iPS cell line is missing.

(Response) We contacted the PBD project (the organization that shared PBD cell line) – we are the first paper to publish these patient cells. Cell line creation and confirmation was done by the Genome Engineering & Stem Cell Center (GESCC) at Washington University in St. Louis, Baylor College of Medicine, and Rarebase, PBC, this information is added to the acknowledgements.

Line 148: There should be a period after "... import into peroxisomes." Also, the next sentence would be simpler if it were written "In PEX10 mutant cells, import of GFP-SKL was delayed after...".

(Response) Thank you, corrected.

Line 372: In the legend to the quantitation in Fig. 3C, the label should be "(c)" instead of "(d)".

(Response) Thank you, sorry for the mistake, corrected.

Reviewer #2 (Remarks to the Author)

I have looked at the construction of the paper and the results and I find them noteworthy. The reports on the live-cell imaging of peroxisomes are not very common and authors demonstrate clean results. Peroxisome imaging is achieved by Bodipy-C12 and siR-C18 derivative and the images are of high quality.

(Response) We thank the reviewer for their supportive and interesting comments. We apologize for a delayed response; it was due to my lab relocation to the University of Southampton and a delay in obtaining additional patient cell lines. In the revised manuscript we tested additional peroxisomal dyes and some of the experiments were replaced with an improved version of the far-red peroxisomal dye – PeroxiSPY650 (instead of SiR-C18). You can find the dye comparisons, toxicity tests in Figures S5a-b, and S6 in the revised version of the manuscript.

The fluorophores with hydrophobic tails might get incorporated in the plasma membrane or gets into the lipid droplets. So it is interesting that these probes gets into peroxisomes as opposed to LDs. The authors do mention its use for LD in the first sentence with a few citations and others ([10.1371/journal.pone.0153522](https://doi.org/10.1371/journal.pone.0153522); [https://elifesciences.org/articles/66393](https://doi.org/10.1017/S1431927620016761); [10.1017/S1431927620016761](https://doi.org/10.1017/S1431927620016761)), but I am unclear what contributed to those specks being observed. What contributes to these changes vis-a-vis fixed vs live cells?

(Response) We think that the dyes could be incorporated into peroxisomes by ABC transporters, for example in the CRISPR/Cas9 KO of the major ABC transporter – PMP70, the BODIPY-C12 staining of peroxisomes is abolished, and PeroxiSPY is significantly reduced (see Figure 4a-e). ABC transporters require energy, that would explain no staining in the fixed cells. However, at this point we do not have enough evidence to prove the direct transport hypothesis, we will continue work on the mechanism in the follow up manuscript. Another hypothesis, suggested by Reviewer 1, is that the properties of the membrane in PMP70 KO change and that leads to a reduced incorporation of the dye.

Interestingly, we found that hydrophobic tails alone are not sufficient to stain lipid droplets – as SiR dyes fused to C16, C18, or C20 fatty acids did not localize to lipid droplets (Figure 2b-c, compare to BODIPY-based dyes Figure 2d-e). However as this reviewer mentioned BODIPY is lipophilic and in a longer incubation time (e.g. 30min) the signal gradually develops in the Lipid droplets (see Figure 2d-e). Interestingly peroxisomes are stained with BODIPY-C12 before lipid droplets, during fixation the active transport would be abolished – which will prevent staining of peroxisomes, however it will not prevent passive diffusion of the dye into lipophilic compartments – BODIPY-C12 can be used in fixed cells as well, where it will stain lipid droplets.

Are they motile and do authors observe interaction with other organelle after sometime?

(Response) These dyes can be used to study motility of peroxisomes, we attach the movie where peroxisome dynamics is visualized during an hour long acquisition in live cells. Peroxisomes exhibit low range of movements, with few compartments moving faster than others which can be traced using tracking software (e.f. ImageJ TrackMate plugin). The dye signal is co-localizing with the classical peroxisome import marker (GFP-SKL) throughout the acquisition (see Supplementary movie 1). Although we did not measure interaction of peroxisomes with other compartments, we are preparing a follow up manuscript describing that dye exit from peroxisome is regulated by contact sites with ER.

After how much time or until what time, these probes localized themselves in peroxisomes. Do they remain so?

(Response) The new probes that we synthesized are more stable in peroxisomes and remain in them for 2 hours (in cell lines tested). The probes appear in peroxisomes after 5min and then increase to a steady intensity level after 10min after staining. BODIPY-C12 probe will also appear in lipid droplets after 15 min of incubation. If the concentration of the dye exceeds 2uM we noticed a gradual increase in the background level in addition to the peroxisomal staining (see Figure 2f-j in which we test the displacement of BODIPY by PeroxiSPY).

I think this is a good study for live-cell imaging of peroxisomes.

(Response) Thank you, we really appreciate that.

Reviewer #3 (Remarks to the Author)

The manuscript of Daria Korotkova et al entitled “A new fluorescent fatty acid conjugate for live cell imaging of peroxisomes” is an interesting work concerning the application of Bodipy-C12 as a fluorescence probe for functional and dysfunctional peroxisomes in live cells. Due to the important and innovating results, the manuscript could be accepted for publication but only after a major revision concerning some aspects:

(Response) We thank this Reviewer for their constructive comments. We addressed them below. We apologize for the delayed response. That was due to my lab relocation to the University of Southampton and delay in obtaining the patient cells required for the revision. Please note that in the revised version we tested more peroxisomal dyes and some of the experiments were replaced with an improved version of the far-red peroxisomal dye – PeroxiSPY650 (instead of SiR-C18). You can find the dye comparisons, toxicity tests, and displacement experiments in Figures S5a-b, S6 (toxicity), and Figure 2 (specificity and displacement). We renamed the selected dyes as PeroxiSPY650 and PeroxiSPY555.

1- In the methods' section of microscopy, the authors should attribute each laser to the correspondent fluorescent dye and the emission band recorded for each dye (line 286 and 287).

(Response) We added the information to the methods (Microscopy section) for all the peroxisome dyes that we used in this study.

2- In figure 1(b), is the intensity profile along a single peroxisome? In order to be a more representative analysis of the dye's intracellular distribution, the intensity profile should be represented along an entire cell.

(Response) We added the intensity along the entire cell (Figure S1d).

3- The PCC (Pearson's correlation coefficient) should be determined to quantify the colocalization images of the manuscript.

(Response) Thank you for pointing this out. We added the Pearson correlation coefficient to all the co-localization panels – in the legends and in certain cases in the Figure panels.

4- Is the intensity profile in figure 3 (a) representing the fluorescence pattern of Bodipy-C12 in WT and PEX5 KO HEK293T cells? It's not very clear. This type of plots, normally, are applied to represent the fluorescence intensity of two or more dyes in order to demonstrate the presence/absence of colocalization along the same spatial region (same image). In this case, the authors should plot the intensity profiles of Bodipy-C12 and GFP-SKL per cell line: one plot of the WT cells and other plot of PEX5 KO HEK293T cells.

(Response) Thank you for pointing this out. We replaced the overlaid profiles with a separate profile for WT and PEX5 KO HEK293T cells for both channels as was recommended (Figure 3a). We also included Pearson correlation coefficient as a readout of co-localization.

In figure 4 (f), the intensity profile represents the distribution of the Bodipy-C12 in different spatial regions, what can be concluded from that?

(Response) In the revised manuscript we indicated that these are different spatial regions in the Figure legend (Figure 4f, 4i-j). This overlaid intensity profile graph was meant to help illustrate the difference in peroxisome distribution (fewer peroxisomes are present in the same profile distance). Additionally, an increase in staining intensity is visible. In the follow up manuscript we show that the increase in VAPA/B peroxisomal staining is associated with the reduction in the dye exit from peroxisomes in VAPA/B KO cells.

5- In figure 5 (a), the authors shouldn't give the same name for structurally different BODIPYs (Bodipy-C12).

(Response) Thank you for pointing this out, we corrected it with the BODIPY-FL-C12 (for the green dye).

6- In figure 5 (a), the microscope images of the PEX19 KO HEK293T cells stained with SiR-C18 appear to be wrong: in the SiR-C18 there is no sign of fluorescence, and in the correspondent merged image there is a residual/ background blue fluorescence (right side of the cell's nucleus).

(Response) We corrected the mistaken image, it was the Hoechst channel instead of the dye channel, thank you for pointing this out. We also now replaced all the SiR-C18 experiments with the PeroxiSPY650 data.

Reviewers' Comments:

Reviewer #1:

Remarks to the Author:

The authors have addressed all of our initial specific concerns. Nevertheless, we feel that the revised manuscript has a couple of major issues. First, the authors add new data that show that neither BODIPY-C12 nor the optimized peroxisome probes work in plants; this information really weakens the utility of the probes. (Could it be that the failure of the probes to stain peroxisomes in plants reflects a technical issue that could be optimized, rather than a biological property of the model system?) In addition, these data contrast with those in a previous report that demonstrated peroxisome staining of a BODIPY-based dye that only worked in plants. There is no rationale provided for this discrepancy.

Second, the new organization of the abstract and the paper weakens the novelty and significance of the findings. Previously, the authors first highlighted the unexpected observation that BODIPY-C12 not only stains lipid droplets but also peroxisomes, then described the probe's staining characteristics, and finally introduced a new probe that was shown to be superior in its staining behavior. In the new version, the authors jump back and forth between BODIPY-C12 and the optimized dyes, and so the superiority of the new probes doesn't come across that well. It is also confusing that in some contexts (e.g., zebrafish embryos), BODIPY-C12 actually performs better than the optimized probes. We feel that the impact of the paper would be greatly improved if it focused on mammalian cells from the beginning. The description of BODIPY-C12 and the characterization of its staining mechanism logically should come first, followed by the development of the optimized probes and their comparison to BODIPY-C12, and finally the use of the optimized probes in patient-derived cell lines. Potential applications and limitations of the probes in other systems can be described and discussed at the end.

Third, the mechanism by which the probes stain peroxisomes remains unclear, although this is not the major point of the paper. Clearly the PMP70 protein, an ABC transporter, is not the only protein required for dye uptake.

Finally, the wording of the paper is often clumsy and there are a number of grammatical mistakes that need to be corrected.

Reviewer #2:

Remarks to the Author:

I checked the response of the authors to the suggestions raised by the reviewers and find them satisfactory. They have performed additional studies, added additional molecules and this makes it much stronger than the first draft.

The paper may be accepted.

Reviewer #3:

Remarks to the Author:

The manuscript of Tame et al is a very interesting and complete work concerning the synthesis, characterization and evaluation of novel heterocyclic fluorescent fatty acids (PeroxiSPY dyes), as live-cell imaging of peroxisomes.

Due to the excellent and innovating results, the manuscript should be accepted as it is.

Reviewer #1 (Remarks to the Author):

(Reviewer) The authors have addressed all of our initial specific concerns. Nevertheless, we feel that the revised manuscript has a couple of major issues.

(Response) Dear Reviewer 1, thank you for the constructive criticism that has significantly improved our work. We highlighted the changes in the revised text.

(Reviewer) First, the authors add new data that show that neither BODIPY-C12 nor the optimized peroxisome probes work in plants; this information really weakens the utility of the probes. (Could it be that the failure of the probes to stain peroxisomes in plants reflects a technical issue that could be optimized, rather than a biological property of the model system?) In addition, these data contrast with those in a previous report that demonstrated peroxisome staining of a BODIPY-based dye that only worked in plants. There is no rationale provided for this discrepancy.

(Response) The BODIPY-based plant peroxisome probes from Landrum *et al.* are difficult to compare to our dyes chemically. Apart from the BODIPY core, there are no similarities. In our study we also replaced the BODIPY part in PeroxiSPY650 and improved specificity of peroxisome staining, indicating that the BODIPY core is not responsible for the staining in mammalian cells. Please see the chemical structures below:

Figure for Reviewer.

- (a) Chemical structure of the plant BODIPY-based probes adapted from Figure 1 from Landrum *et al.*, 2010.
- (b) BODIPY-C12 and PeroxiSPY probes from this study

(Reviewer) (Could it be that the failure of the probes to stain peroxisomes in plants reflects a technical issue that could be optimized, rather than a biological property of the model system?) We collaborated with Dr. Anthony Guihur – plant scientist with a significant expertise in plant culture and microscopic visualization of plant cells to avoid this issue. We considered both the cell wall retention problem (by using spheroplasts) and chloroplast fluorescence overlap (by using roots). Overall, the technical issue is very unlikely. It is however possible that in other plants (not *Arabidopsis*) the results would differ, therefore we described the limitations as applicable to the system tested (*A. thaliana* seedlings and spheroplasts). Overall, we think that the limitations of the dye application are informative and should be reported in full, however we agree with the reviewer on the place and emphasis of

this data (see next comment).

(Reviewer) Second, the new organization of the abstract and the paper weakens the novelty and significance of the findings. Previously, the authors first highlighted the unexpected observation that BODIPY-C12 not only stains lipid droplets but also peroxisomes, then described the probe's staining characteristics, and finally introduced a new probe that was shown to be superior in its staining behavior. In the new version, the authors jump back and forth between BODIPY-C12 and the optimized dyes, and so the superiority of the new probes doesn't come across that well. It is also confusing that in some contexts (e.g., zebrafish embryos), BODIPY-C12 actually performs better than the optimized probes. We feel that the impact of the paper would be greatly improved if it focused on mammalian cells from the beginning. The description of BODIPY-C12 and the characterization of its staining mechanism logically should come first, followed by the development of the optimized probes and their comparison to BODIPY-C12, and finally the use of the optimized probes in patient-derived cell lines. Potential applications and limitations of the probes in other systems can be described and discussed at the end.

(Response) We agree with the reviewer. We revised the structure of the abstract, returning largely to the previous version. We also restructured the manuscript in the way suggested by this reviewer – focusing on mammalian cells, mentioning the potential applications and limitations in the end in the supplementary figures, and following the suggested figure sequence. We think that mechanistic details (e.g. PMP70 data) are important and will serve as a starting point for the funding application to develop this project further.

(Reviewer) Third, the mechanism by which the probes stain peroxisomes remains unclear, although this is not the major point of the paper. Clearly the PMP70 protein, an ABC transporter, is not the only protein required for dye uptake.

(Response) This is a very interesting point that we tried to address for this revision however after considering all of the data we decided to prepare a separate manuscript in which we will thoroughly compare all the ABC transporters (ABCD1,2,3) using CRISPR/Cas9 knockouts which will take time to construct. We contacted the lab that has the transporter KO available with a collaboration proposal, but we did not receive a response. In addition to that, deciphering the mechanism will require answering if the dye is localized to the lumen or the membrane, for that we will establish a super resolution microscopy approach. As this reviewer pointed out, the mechanism was not the point of the paper, however we wanted to characterize how major transporters affect the dye import which emphasizes that even similarly constructed dyes (fatty acid fusion to a fluorophore) can have different import dependencies.

(Reviewer) Finally, the wording of the paper is often clumsy and there are a number of grammatical mistakes that need to be corrected.

(Response) We apologize for the mistakes. We showed the manuscript to a native speaker to improve the writing and corrected the grammatical mistakes to the best of our ability. Please see the revised version of the paper.

Reviewer #2 (Remarks to the Author):

(Reviewer) I checked the response of the authors to the suggestions raised by the reviewers and find them satisfactory. They have performed additional studies, added additional molecules and this makes it much stronger than the first draft.

The paper may be accepted.

(Response) We thank this reviewer for their acceptance of our work for publication.

Reviewer #3 (Remarks to the Author):

(Reviewer) The manuscript of Tame et al is a very interesting and complete work concerning the synthesis, characterization and evaluation of novel heterocyclic fluorescent fatty acids (PeroxiSPY dyes), as live-cell imaging of peroxisomes.

Due to the excellent and innovating results, the manuscript should be accepted as it is.

(Response) We thank this reviewer for their positive feedback on our work.

Reviewers' Comments:

Reviewer #1:

Remarks to the Author:

Overall, the revised manuscript reads much better, and we have no further issues before publication except some grammatical improvements (see below) and re-wording the second paragraph of the discussion. This paragraph is much too long and discursive and should be broken down into two smaller paragraphs. One of these should discuss the dye import mechanism shown in Figure 5i (which is never explained in the text or in the figure legend) and comment on why the duration of peroxisomal staining would be affected by matrix protein import (perhaps the dyes are normally bound inside peroxisomes by certain matrix proteins); why staining would be stronger in the absence of contact sites (which is illustrated in the model by bi-directional lipid transfer but never explained in the text); and also why different cell types such as fibroblasts can only be stained by some of the dyes (e.g., differential transporter expression as the authors hypothesize). The utility of the dyes to reveal peroxisomal dysfunctions in vitro and in disease settings can then be the focus of another separate paragraph.

Minor grammatical suggestions:

Lines 54-55: It would be helpful to highlight the connection to patient samples in this sentence, perhaps rewording it as follows: "...approaches are, therefore, inadequate for investigating peroxisome dysfunction and peroxisome biogenesis abnormalities, in particular those that may occur in patient samples."

Lines 112-113: The sentence could be clarified to read "We further focused on the best performing dyes, SiR-C20 and SPY555-C20, which we will refer to as PeroxiSPY650 and PeroxiSPY555, respectively."

Line 126: The sentence could be clarified to read "Peroxisomes are usually visualized with the import marker GFP-SKL, which is imported into peroxisomes..."

Line 128: The sentence could be clarified to read "BODIPY-C12 stained peroxisomes independently of peroxisomal function in PEX5 KO cells."

Lines 128-129: The sentence could be clarified to read " We noticed, however, that staining in PEX5 KO cells is transient..."

Lines 134-136: The sentence could be clarified to read "We also noticed that the timing of BODIPY-C12 staining in WT cells inversely depended on the dye concentration (Figure S5d): higher concentrations led to an overload..."

Lines 150-151: The sentence could be clarified to read "...indicating that contact sites unlikely mediate the import of the dyes into peroxisomes."

Lines 151-153: The sentence could be broken up and clarified to read "These results suggest instead the possibility of substrate-like import by ATP-dependent transporters. Indeed, peroxisomes in fixed cells were stained neither by the BODIPY C12-based dyes nor by our optimized probes (Figure 4k)."

Line 155: The sentence could be clarified to read "To determine whether peroxisome staining could identify peroxisome dysfunctions in disease contexts, we..."

Line 195: The sentence could be clarified to read "...also synthesized highly specific far-red and red photostable peroxisomal dyes that we collectively call PeroxiSPY."

Lines 198-200: The sentence could be clarified to read "Our proof-of-concept data in mammalian cells demonstrate that accumulation of the dyes in peroxisomes depends on peroxisomal function and is sensitive to minor changes in peroxisomal matrix protein import rates..."

Line 202: The sentence could be clarified to read "...utilized PBD patient-derived iPSCs and..."

Lines 203-204: The sentence could be clarified to read "Both BODIPY-C12 and PeroxiSPY probes exhibited differential staining of functional and dysfunctional peroxisomes: PeroxiSPY..."

Reviewer #1 (Remarks to the Author):

Overall, the revised manuscript reads much better, and we have no further issues before publication except some grammatical improvements (see below) and re-wording the second paragraph of the discussion. This paragraph is much too long and discursive and should be broken down into two smaller paragraphs. One of these should discuss the dye import mechanism shown in Figure 5i (which is never explained in the text or in the figure legend) and comment on why the duration of peroxisomal staining would be affected by matrix protein import (perhaps the dyes are normally bound inside peroxisomes by certain matrix proteins); why staining would be stronger in the absence of contact sites (which is illustrated in the model by bi-directional lipid transfer but never explained in the text); and also why different cell types such as fibroblasts can only be stained by some of the dyes (e.g., differential transporter expression as the authors hypothesize). The utility of the dyes to reveal peroxisomal dysfunctions in vitro and in disease settings can then be the focus of another separate paragraph.

[Response] We sincerely thank this reviewer for all their comments that improved our manuscript. We revised the discussion according to the reviewer suggestions.

Reviewer #1 Minor grammatical suggestions:

Lines 54-55: It would be helpful to highlight the connection to patient samples in this sentence, perhaps rewording it as follows: "...approaches are, therefore, inadequate for investigating peroxisome dysfunction and peroxisome biogenesis abnormalities, in particular those that may occur in patient samples."

Lines 112-113: The sentence could be clarified to read "We further focused on the best performing dyes, SiR-C20 and SPY555-C20, which we will refer to as PeroxiSPY650 and PeroxiSPY555, respectively."

Line 126: The sentence could be clarified to read "Peroxisomes are usually visualized with the import marker GFP-SKL, which is imported into peroxisomes..."

Line 128: The sentence could be clarified to read "BODIPY-C12 stained peroxisomes independently of peroxisomal function in PEX5 KO cells."

Lines 128-129: The sentence could be clarified to read " We noticed, however, that staining in PEX5 KO cells is transient..."

Lines 134-136: The sentence could be clarified to read "We also noticed that the timing of BODIPY-C12 staining in WT cells inversely depended on the dye concentration (Figure S5d): higher concentrations led to an overload..."

Lines 150-151: The sentence could be clarified to read "...indicating that contact sites unlikely mediate the import of the dyes into peroxisomes."

Lines 151-153: The sentence could be broken up and clarified to read "These results suggest instead the possibility of substrate-like import by ATP-dependent transporters. Indeed, peroxisomes in fixed cells were stained neither by the BODIPY C12-based dyes nor by our optimized probes (Figure 4k)."

Line 155: The sentence could be clarified to read "To determine whether peroxisome

staining could identify peroxisome dysfunctions in disease contexts, we..."

Line 195: The sentence could be clarified to read "...also synthesized highly specific far-red and red photostable peroxisomal dyes that we collectively call PeroxiSPY."

Lines 198-200: The sentence could be clarified to read "Our proof-of-concept data in mammalian cells demonstrate that accumulation of the dyes in peroxisomes depends on peroxisomal function and is sensitive to minor changes in peroxisomal matrix protein import rates..."

Line 202: The sentence could be clarified to read "...utilized PBD patient-derived iPSCs and..."

Lines 203-204: The sentence could be clarified to read "Both BODIPY-C12 and PeroxiSPY probes exhibited differential staining of functional and dysfunctional peroxisomes: PeroxiSPY..."

[Response] We corrected all the suggested sentences.